# A phylogenetic transform enhances analysis of compositional microbiota data

**Justin D Silverman**[1,2,3], **Alex D Washburne**[4,5]**, Sayan Mukherjee**[1,6,7,8,9]**, Lawrence A David**[1,3,10]*****

[1]Program in Computational Biology and Bioinformatics, Duke University, Durham, United States; [2]Medical Scientist Training Program, Duke University, Durham, United States; [3]Center for Genomic and Computational Biology, Duke University, Durham, United States; [4]Nicholas School of the Environment, Duke University, Durham, United States; [5]Cooperative Institute for Research in Environmental Sciences (CIRES), University of Colorado, Boulder, United States; [6]Department of Statistical Science, Duke University, Durham, United States; [7]Department of Mathematics, Duke University, Durham, United States; [8]Department of Biostatistics and Bioinformatics, Duke University, Durham, United States; [9]Department of Computer Science, Duke University, Durham, United States; [10]Department of Molecular Genetics and Microbiology, Duke University, Durham, United States

**Abstract** Surveys of microbial communities (microbiota), typically measured as relative abundance of species, have illustrated the importance of these communities in human health and disease. Yet, statistical artifacts commonly plague the analysis of relative abundance data. Here, we introduce the PhILR transform, which incorporates microbial evolutionary models with the isometric log-ratio transform to allow off-the-shelf statistical tools to be safely applied to microbiota surveys. We demonstrate that analyses of community-level structure can be applied to PhILR transformed data with performance on benchmarks rivaling or surpassing standard tools. Additionally, by decomposing distance in the PhILR transformed space, we identified neighboring clades that may have adapted to distinct human body sites. Decomposing variance revealed that covariation of bacterial clades within human body sites increases with phylogenetic relatedness. Together, these findings illustrate how the PhILR transform combines statistical and phylogenetic models to overcome compositional data challenges and enable evolutionary insights relevant to microbial communities.

**\*For correspondence:** lawrence. david@duke.edu

**Competing interests:** The authors declare that no competing interests exist.

## Introduction

Microbiota research today embodies the data-rich nature of modern biology. Advances in high-throughput DNA sequencing allow for rapid and affordable surveys of thousands of bacterial taxa across hundreds of samples (*Caporaso et al., 2011*). The exploding availability of sequencing data has poised microbiota research to advance our understanding of fields as diverse as ecology, evolution, medicine, and agriculture (*Waldor et al., 2015*). Considerable effort now focuses on interrogating microbiota datasets to identify relationships between bacterial taxa, as well as between microbes and their environment.

Increasingly, it is appreciated that the relative nature of microbial abundance data in microbiota studies can lead to spurious statistical analyses (*Jackson, 1997*; *Friedman and Alm, 2012*; *Aitchison, 1986*; *Lovell et al., 2011*; *Gloor et al., 2016a*; *Britanova et al., 2014*; *Li, 2015*; *Tsilimigras and Fodor, 2016*). With next generation sequencing, the number of reads per sample

can vary independently of microbial load (*Lovell et al., 2011*; *Tsilimigras and Fodor, 2016*). In order to make measurements comparable across samples, most studies therefore analyze the relative abundance of bacterial taxa. Analyses are thus not carried out on absolute abundances of community members (*Figure 1A*), but rather on relative data occupying a constrained geometric space and represented in a non-Cartesian coordinate system (*Figure 1B*). Such relative abundance datasets are often termed compositional. The use of most standard statistical tools (*e.g.*, correlation, regression, or classification) within a compositional space leads to spurious results (*Pawlowsky-Glahn et al., 2015*). For example, three-quarters of the significant bacterial interactions inferred by Pearson correlation on a compositional human microbiota dataset were likely false (*Friedman and Alm, 2012*), and over two-thirds of differentially abundant taxa inferred by a t-test on a simulated compositional human microbiota dataset were spurious (*Mandal et al., 2015*). To account for compositional effects in microbial datasets, bioinformatics efforts have re-derived common statistical methods including correlation statistics (*Friedman and Alm, 2012*; *Fang et al., 2015*), hypothesis testing (*La Rosa et al., 2012*), and variable selection (*Chen and Li, 2013*; *Lin et al., 2014*).

An alternative approach is to transform compositional microbiota data to a space where existing statistical methods may be applied without introducing spurious conclusions. This approach is attractive because of its efficiency: the vast toolbox of existing statistical models can be applied without re-derivation. Normalization methods, for example, have been proposed to modify count data by

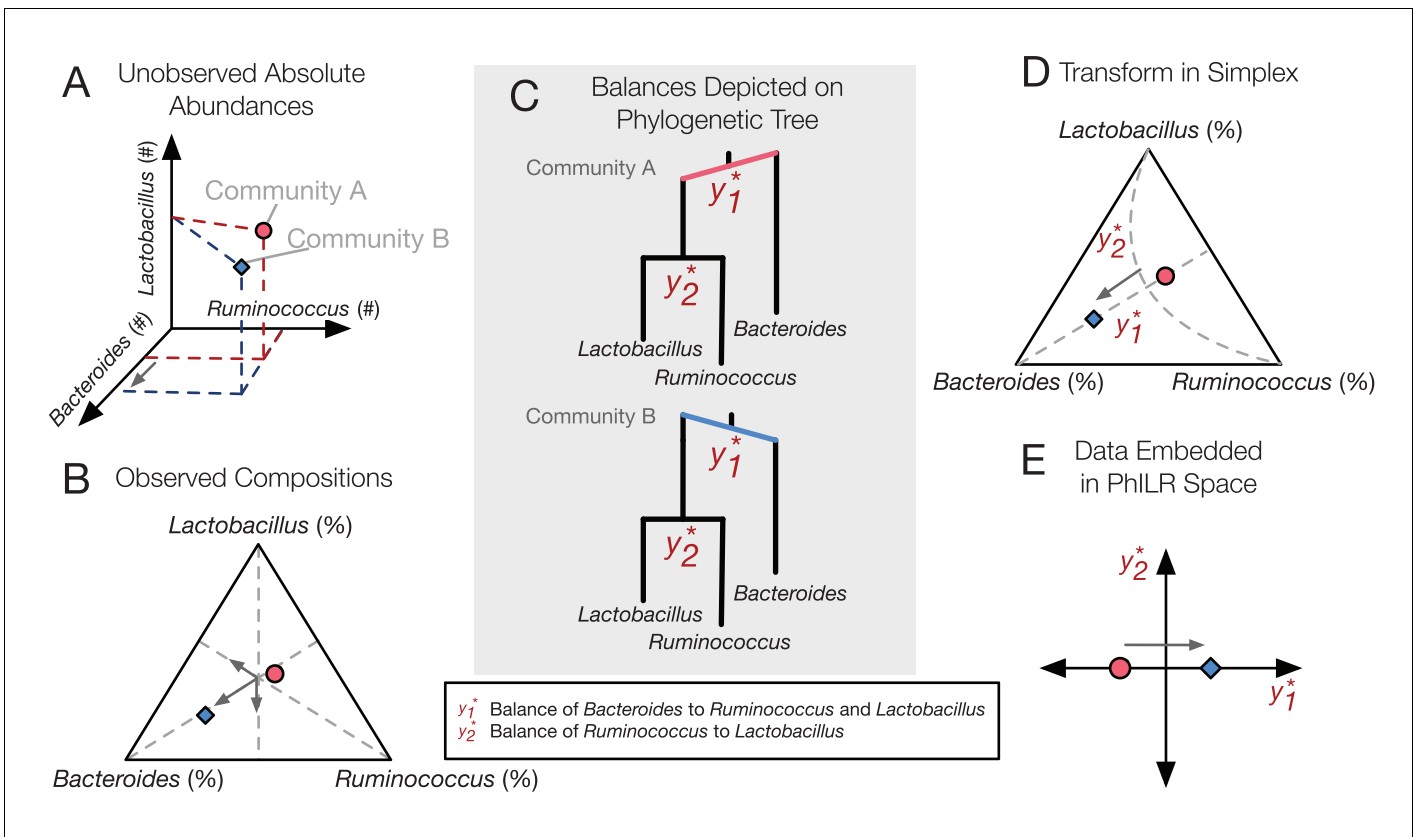

**Figure 1.** PhILR uses an evolutionary tree to transform microbiota data into an unconstrained coordinate system. (**A**) Two hypothetical bacterial communities share identical absolute numbers of *Lactobacillus*, and *Ruminococcus* bacteria; they differ only in the absolute abundance of *Bacteroides* which is higher in community A (red circle) compared to community B (blue diamond). (**B**) A ternary plot depicts proportional data typically analyzed in a sequencing-based microbiota survey. Note that viewed in terms of proportions the space is constrained and the axes are not Cartesian. As a result, all three genera have changed in relative abundance between the two communities. (**C**) Schematic of the PhILR transform based on a phylogenetic sequential binary partition. The PhILR coordinates can be viewed as 'balances' between the weights (relative abundances) of the two subclades of a given internal node. In community B, the greater abundance of *Bacteroides* tips the balance $y_1^*$ to the right. (**D**) The PhILR transform can be viewed as a new coordinate system (grey dashed lines) in the proportional data space. (**E**) The data transformed to the PhILR space. Note that in contrast to the raw proportional data (**B**), the PhILR space only shows a change in the variable associated with *Bacteroides*.

assuming reads follow certain statistical distributions (*e.g.*, negative binomial) (*Paulson et al., 2013*; *Anders and Huber, 2010*). Alternatively, the field of Compositional Data Analysis (CoDA) has focused on formalizing methods for transforming compositional data into a simpler geometry without having to assume data adhere to a distribution model (*Bacon-Shone, 2011*). Previous microbiota analyses have already leveraged CoDA theory and used the centered log-ratio transform to reconstruct microbial association networks and interactions (*Kurtz et al., 2015*; *Lee et al., 2014*) and to analyze differential abundances (*Fernandes et al., 2014*; *Gloor et al., 2016b*). However, the centered log-ratio transform has a crucial limitation: it yields a coordinate system featuring a singular covariance matrix and is thus unsuitable for many common statistical models (*Pawlowsky-Glahn et al., 2015*). This drawback can be sidestepped using another CoDA transform, known as the Isometric Log-Ratio (ILR) transformation (*Egozcue et al., 2003*). The ILR transform can be built from a sequential binary partition of the original variable space (*Figure 1C*), creating a new coordinate system with an orthonormal basis (*Figure 1D and E*) (*Egozcue and Pawlowsky-Glahn, 2005*). However, a known obstacle to using the ILR transform is the choice of partition such that the resulting coordinates are meaningful (*Pawlowsky-Glahn et al., 2015*). To date, microbiota studies have chosen ILR coordinates using ad hoc sequential binary partitions of bacterial groups that are not easily interpreted (*Finucane et al., 2014*; *Lê Cao et al., 2016*). Alternatively, external covariates have been used to pick groups of bacterial taxa to contrast (*Morton et al., 2017*).

Here, we introduce the bacterial phylogenetic tree as a natural and informative sequential binary partition when applying the ILR transform to microbiota datasets (*Figure 1C*). Using phylogenies to construct the ILR transform results in an ILR coordinate system capturing evolutionary relationships between neighboring bacterial groups (clades). Analyses of neighboring clades offer the opportunity for biological insight: clade analyses have linked genetic differentiation to ecological adaptation (*Hunt et al., 2008*), and the relative levels of sister bacterial genera differentiate human cohorts by diet, geography, and culture (*De Filippo et al., 2010*; *Wu et al., 2011*; *Yatsunenko et al., 2012*). Datasets analyzed by a phylogenetically aware ILR transform could therefore reveal ecological and evolutionary factors shaping host-associated microbial communities.

We term our approach the **Ph**ylogenetic **ILR** (PhILR) transform. Using published environmental and human-associated 16S rRNA datasets as benchmarks, we found that simple Euclidean distances calculated on PhILR transformed data provided a compositionally robust alternative to distance/dissimilarity measures like Bray-Curtis, Jaccard, and Unifrac. In addition, we observed that the accuracy of supervised classification methods on our benchmark datasets was matched or improved with PhILR transformed data relative to applying the same models on untransformed (raw) or log transformed relative abundance data. Decomposing distances between samples along PhILR coordinates identified bacterial clades that may have differentiated to adapt to distinct body sites. Similar decomposition of variance along PhILR coordinates showed that, in all human body sites studied, the degree to which neighboring bacterial clades covary tends to increase with the phylogenetic relatedness between clades. Together, these findings demonstrate that the PhILR transform can be used to enhance existing microbiota analysis pipelines, as well as enable novel phylogenetic analyses of microbial ecosystems.

## Results

### Constructing the PhILR transform

The PhILR transform has two goals. The first goal is to transform input microbiota data into an unconstrained space with an orthogonal basis while preserving all information contained in the original composition. The second goal is to conduct this transform using phylogenetic information. To achieve these dual goals on a given set of $N$ samples consisting of relative measurements of $D$ taxa (*Figure 1B*), we transform data into a new space of $N$ samples and $(D-1)$ coordinates termed 'balances' (*Figure 1C–E*) (*Egozcue et al., 2003*; *Egozcue and Pawlowsky-Glahn, 2005*). Each balance $y_i^*$ is associated with a single internal node $i$ of a phylogenetic tree with the $D$ taxa as leaves (the asterisk denotes a quantity represented in PhILR space). The balance represents the log-ratio of the geometric mean relative abundance of the two clades of taxa that descend from $i$ (Materials and methods). Although individual balances may share overlapping sets of leaves and thus exhibit dependent behavior, the ILR transform rescales and combines leaves to form a coordinate system

whose basis vectors are orthonormal and the corresponding coordinates are Cartesian (*Egozcue et al., 2003*; *Egozcue and Pawlowsky-Glahn, 2005*). The orthogonality of basis vectors allows conventional statistical tools to be used without compositional artifacts. The unit-length of basis vectors makes balances across the tree statistically comparable even when they have differing numbers of descendant tips or exist at different depths in the tree (*Pawlowsky-Glahn et al., 2015*). In addition, the unit-length ensures that the variance of PhILR balances has a consistent scale, unlike the variance of log-ratios originally proposed by Aitchison (*Aitchison, 1986*) as a measure of association, in which it can be unclear what constitutes a large or small variance (*Friedman and Alm, 2012*).

While the above description represents the core of the PhILR transform, we have also equipped the PhILR transform with two sets of weights that can: (1) address the multitude of zero and near-zero counts present in microbiota data; and, (2) incorporate phylogenetic branch lengths into the transformed space. Because zero counts cause problems when computing logs or performing division, zeros are often replaced in microbiota analyses with small non-zero counts. However, to avoid excess zero replacement that may itself introduce bias, stringent hard filtering thresholds are often employed (*e.g.* removing all taxa that are not seen with at least a minimum number of counts in a subset of samples). Still, hard filtering thresholds may remove a substantial fraction of observed taxa and do not account for the low precision (or high variability) of near-zero counts (*Gloor et al., 2016a*; *Good, 1956*; *McMurdie and Holmes, 2014*). We therefore developed a 'taxon weighting' scheme that acts as a type of soft-thresholding, supplementing zero replacement methods with a generalized form of the ILR transform that allows weights to be attached to individual taxa (*Egozcue and Pawlowsky-Glahn, 2016*). Weights are chosen with a heuristic designed to down weight the influence of taxa with many zero or near-zero counts (Materials and methods).

Our second weighting scheme is called branch length weighting. Certain analyses may benefit from incorporating information on evolutionary distances between taxa (*Lozupone and Knight, 2005*; *Fukuyama et al., 2012*; *Purdom, 2011*). For example, because related bacteria may be more likely to share similar traits (*Martiny et al., 2015*), it may be desirable to consider communities differing only in the abundance of closely-related microbes to be more similar than communities differing only in the abundance of distantly-related microbes. Because of the one-to-one correspondence between PhILR balances and internal nodes on the phylogenetic tree, evolutionary information can be incorporated into the PhILR transform by scaling balances using the phylogenetic distance between their direct descendants (Materials and methods). We note that we employ both branch length weighting and taxon weighting throughout our following analyses except where noted; still, these weights should be considered optional additions to the core PhILR transform.

## Benchmarking community-level analyses in the PhILR coordinate system

To illustrate how the PhILR transform can be used to perform standard community-level analyses of microbiota datasets, we first examined measures of community dissimilarity. Microbiota analyses commonly compute the dissimilarity or distance between pairs of samples and use these computed pairwise distances as input to a variety of statistical tools. We investigated how Euclidean distances calculated on PhILR transformed data compared to common ecological measures of microbiota distance or dissimilarity (UniFrac, Bray-Curtis, and Jaccard) as well as Euclidean distance applied to raw relative abundance data in standard distance-based analysis. We chose three different microbiota surveys as reference datasets: Costello Skin Sites (CSS), a dataset of 357 samples from 12 human skin sites (*Costello et al., 2009*; *Knights et al., 2011*); Human Microbiome Project (HMP), a dataset of 4743 samples from 18 human body sites (*e.g.*, skin, vaginal, oral, and stool) (*Human Microbiome Project Consortium, 2012*); and, Global Patterns (GP), a dataset of 26 samples from nine human or environmental sites (*Caporaso et al., 2011*) (*Supplementary file 1* and *Figure 2—figure supplement 1*).

Distance-based analyses using Euclidean distances computed on PhILR transformed data exhibited performance rivaling common ecological distance or dissimilarity measures. Principal coordinate analyses (PCoA) qualitatively demonstrated separation of body sites using both Euclidean distances on PhILR transformed data (*Figure 2A*) and with several standard distance measures calculated on raw relative abundance data (*Figure 2—figure supplement 2*). To quantitatively compare distance measures, we tested how well habitat information explained variability among distance matrices as measured by the $R^2$ statistic from PERMANOVA (*Chen et al., 2012*). By this metric, the Euclidean distance in the PhILR coordinate system significantly outperformed the five competing distance

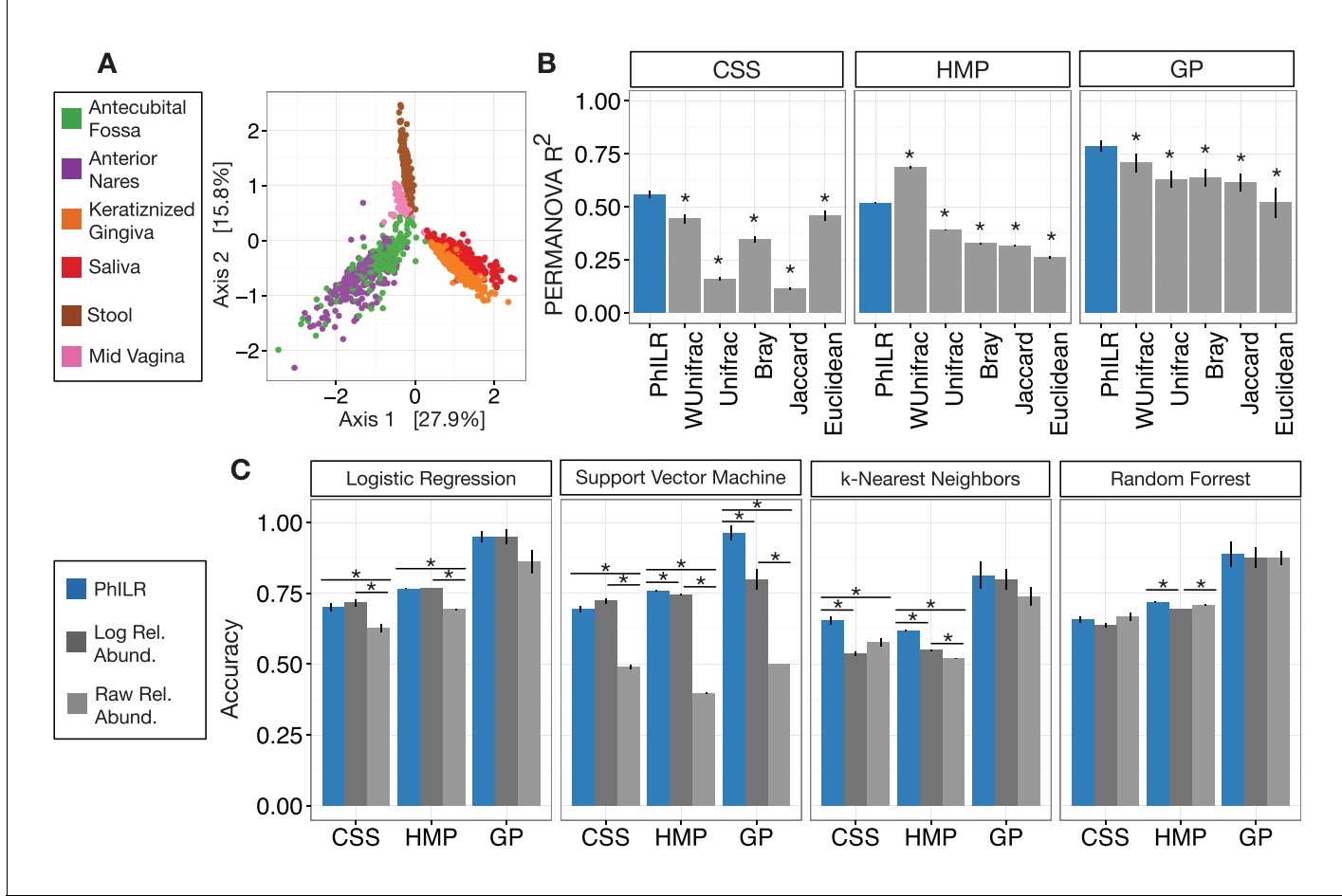

**Figure 2.** Performance of standard statistical models on PhILR transformed microbiota data. Benchmarks were performed using three datasets: Costello Skin Sites (CSS), Global Patterns (GP), Human Microbiome Project (HMP) (a summary of these datasets after preprocessing is shown in *Supplementary file 1* and *Figure 2—figure supplement 1*). (A) Sample distance visualized using principal coordinate analysis (PCoA) of Euclidean distances computed in PhILR coordinate system. A comparison to PCoAs calculated with other distance measures is shown in *Figure 2—figure supplement 2*. (B) Sample distance (or dissimilarity) was computed by a range of statistics. PERMANOVA $R^2$ values, which represent how well sample identity explained the variability in sample pairwise distances, were used as a performance metric. Distances in the PhILR transformed space were calculated using Euclidean distance. Distances between samples on raw relative abundance data were computed using Weighted and Unweighted UniFrac (WUnifrac and Unifrac, respectively), Bray-Curtis, Binary Jaccard, and Euclidean distance. Error bars represent standard error measurements from 100 bootstrap replicates and (*) denotes a p-value of $\leq 0.01$ after FDR correction of pairwise tests against PhILR. (C) Accuracy of supervised classification methods tested on benchmark datasets. Error bars represent standard error measurements from 10 test/train splits and (*) denotes a p-value of $\leq 0.01$ after FDR correction of all pairwise tests.

The following source data and figure supplements are available for figure 2:

**Source data 1.** Source data for *Figure 2b and c* as well as FDR corrected p-values from tests.

**Figure supplement 1.** Taxa weighting scheme tends to assign smaller weights to taxa with more zero and near zero counts.

**Figure supplement 2.** Principal coordinate analyses using different measures of community distance or dissimilarity.

metrics in all but one case (in comparison to Weighted UniFrac when applied to the HMP dataset; *Figure 2B*).

Next, we tested the performance of predictive statistical models in the PhILR coordinate system. We examined four standard supervised classification techniques: logistic regression (LR), support vector machines (SVM), k-nearest neighbors (kNN), and random forests (RF) (*Knights et al., 2011*).

We applied these methods to the same three reference datasets used in our comparison of distance metrics. As a baseline, the machine learning methods were applied to raw relative abundance datasets and raw relative abundance data that had been log-transformed.

The PhILR transform significantly improved supervised classification accuracy in 7 of the 12 benchmark tasks compared to raw relative abundances (*Figure 2C*). Accuracy improved by more than 90% in two benchmarks (SVM on HMP and GP), relative to results on the raw data. Log transformation of the data also improved classifier accuracy significantly on 6 of the 12 benchmarks but also significantly underperformed on one benchmark compared to raw relative abundances. In addition, the PhILR transform significantly improved classification accuracy in 5 of the 12 benchmarks relative to the log transform. Overall, the PhILR transform often outperformed the raw and log transformed relative abundances with respect to classification accuracy and was never significantly worse.

## Identifying neighboring clades that differ by body site preference

While our benchmarking experiments demonstrated how PhILR transformed data performed in community-level analyses, we also wanted to explore potential biological insights afforded by the PhILR coordinate system. We therefore investigated how distances decomposed along PhILR balances using a sparse logistic regression model to examine which balances distinguished human body site microbiota in the HMP dataset. Such balances could be used to identify neighboring bacterial clades whose relative abundances capture community-level differences between body site microbiota. Microbial genetic differentiation may be associated with specialization to new resources or lifestyle preferences (*Hunt et al., 2008*), meaning that distinguishing balances near the tips of the bacterial tree may correspond to clades adapting to human body site environments.

We identified dozens of highly discriminatory balances, which were spread across the bacterial phylogeny (*Figure 3A* and *Figure 3—figure supplement 1*). Some discriminatory balances were found deep in the tree. Abundances of the Firmicutes, Bacteroidetes, and Proteobacteria relative to the Actinobacteria, Fusobacteria, and members of other phyla, separated skin body sites from oral and stool sites (*Figure 3B*). Levels of the genus *Bacteroides* relative to the genus *Prevotella* differentiated stool microbiota from other communities on the body (*Figure 3C*). Notably, values of select balances below the genus level also varied by body site. Relative levels of sister *Corynebacterium* species separated human skin sites from gingival sites (*Figure 3D*). Species-level balances even differentiated sites in nearby habitats; levels of sister *Streptococcus* species or sister *Actinomyces* species vary depending on specific oral sites (*Figure 3E and F*). These results show that the decomposition of distances between groups of samples along PhILR balances can be used to highlight ancestral balances that distinguish body site microbiota, as well as to identify more recent balances that may separate species that have adapted to inhabit different body sites.

## Balance variance and microbiota assembly

As a natural extension of our analysis of how distance decomposes along PhILR balances, we next investigated how balance variance decomposed in the PhILR coordinate system. Balance variance is a measure of association between neighboring bacterial clades. When the variance of a balance between two clades approaches zero, the mean abundance of taxa in each of the two clades will be linearly related and thus covary across microbial habitats (*Lovell et al., 2015*). By contrast, when a balance exhibits high variance, related bacterial clades exhibit unlinked or exclusionary patterns across samples. Unlike standard measures of association (*e.g.*, Pearson correlation) balance variance is robust to compositional effects (*Pawlowsky-Glahn et al., 2015*).

Our preliminary investigation demonstrated a striking pattern in which balance variance decreased for balances closer to the tips of the phylogeny and increased for balances nearer to the root. To determine if this observed pattern was not the result of technical artifact, we took the following three steps. First, we omitted branch length weights from the transform as we anticipated that branch lengths may vary non-randomly as a function of depth in the phylogeny. Second, we anticipated that balances near the tips of the phylogeny would be associated with fewer read counts and thus would be more biased by our chosen heuristics for taxon weighting and zero replacement. We therefore omitted taxon weighting, employed more stringent filtering thresholds, and conditioned our calculation of balance variance on non-zero counts rather than using zero-replacement (Materials and methods). Third, we combined regression and a permutation scheme to test the null

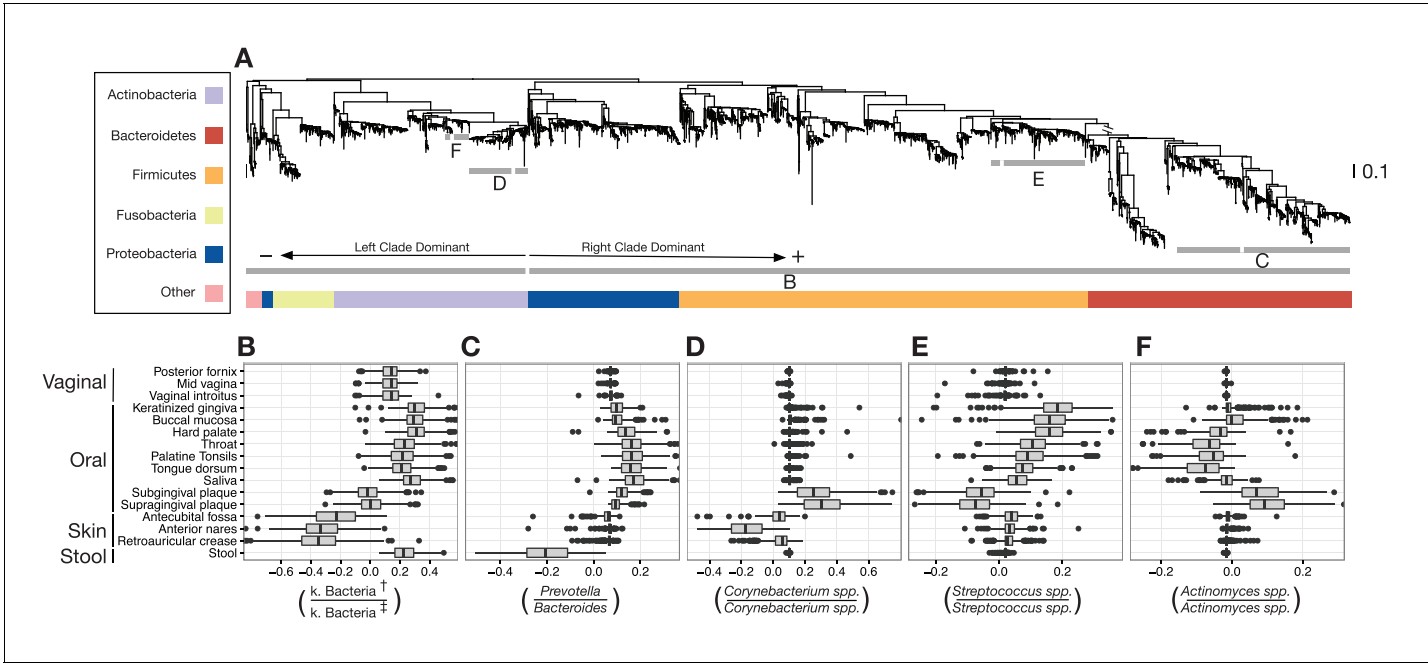

**Figure 3.** Balances distinguishing human microbiota by body site. Sparse logistic regression was used to identify balances that best separated the different sampling sites (full list of balances provided in *Figure 3—figure supplement 1*). (A) Each balance is represented on the tree as a broken grey bar. The left portion of the bar identifies the clade in the denominator of the log-ratio, and the right portion identifies the clade in the numerator of the log-ratio. The branch leading from the Firmicutes to the Bacteroidetes has been rescaled to facilitate visualization. (B–F) The distribution of balance values across body sites. Vertical lines indicate median values, boxes represent interquartile ranges (IQR) and whiskers extend to 1.5 IQR on either side of the median. Balances between: (B) the phyla Actinobacteria and Fusobacteria versus the phyla Bacteroidetes, Firmicutes, and Proteobacteria distinguish stool and oral sites from skin sites; (C) *Prevotella spp.* and *Bacteroides spp.* distinguish stool from oral sites; (D) *Corynebacterium spp.* distinguish skin and oral sites; (E) *Streptococcus spp.* distinguish oral sites; and, (F) *Actinomyces spp.* distinguish oral plaques from other oral sites. (†) Includes Bacteroidetes, Firmicutes, Alpha-, Beta-, and Gamma-proteobacteria. (‡) Includes Actinobacteria, Fusobacteria, Epsilon-proteobacteria, Spirochaetes, and Verrucomicrobia.

The following figure supplement is available for figure 3:

**Figure supplement 1.** Balances found to distinguish human body sites by sparse logistic regression.

hypothesis that the degree to which neighboring clades covary is independent of the phylogenetic distance between them (Materials and methods). By permuting tip labels on the tree, our test generates a restricted subset of random sequential binary partitions that still maintains the count variability (and potential biases due to our zero handling methods) of the observed data.

With our modified PhILR analysis in place, we observed significantly decreasing balance variances near the tips of the phylogenetic tree for all body sites in the HMP dataset ($p<0.01$, permutation test with FDR correction; *Figure 4A–F* and *Figure 4—figure supplements 1–2*). Low variance balances predominated near the leaves of the tree. Examples of such balances involved *B. fragilis* species in stool (*Figure 4H*), *Rothia mucilaginosa* species in the buccal mucosa (*Figure 4J*), and *Lactobacillus* species in the mid-vagina (*Figure 4L*). By contrast, higher variance balances tended to be more basal on the tree. Three examples of high variance balances corresponded with clades at the order (*Figure 4G*), family (*Figure 4I*), and genus (*Figure 4K*) levels. We also observed that the relationship between balance variance and phylogenetic depth varied at different taxonomic scales. LOESS regression revealed that trends between variance and phylogenetic depth were stronger above the species level than below it (Materials and methods; *Figure 4D–F* and *Figure 4—figure supplement 2*). Overall, the observed pattern of decreasing balance variance near the tips of the phylogenetic tree suggested that closely related bacteria tend to covary in human body sites.

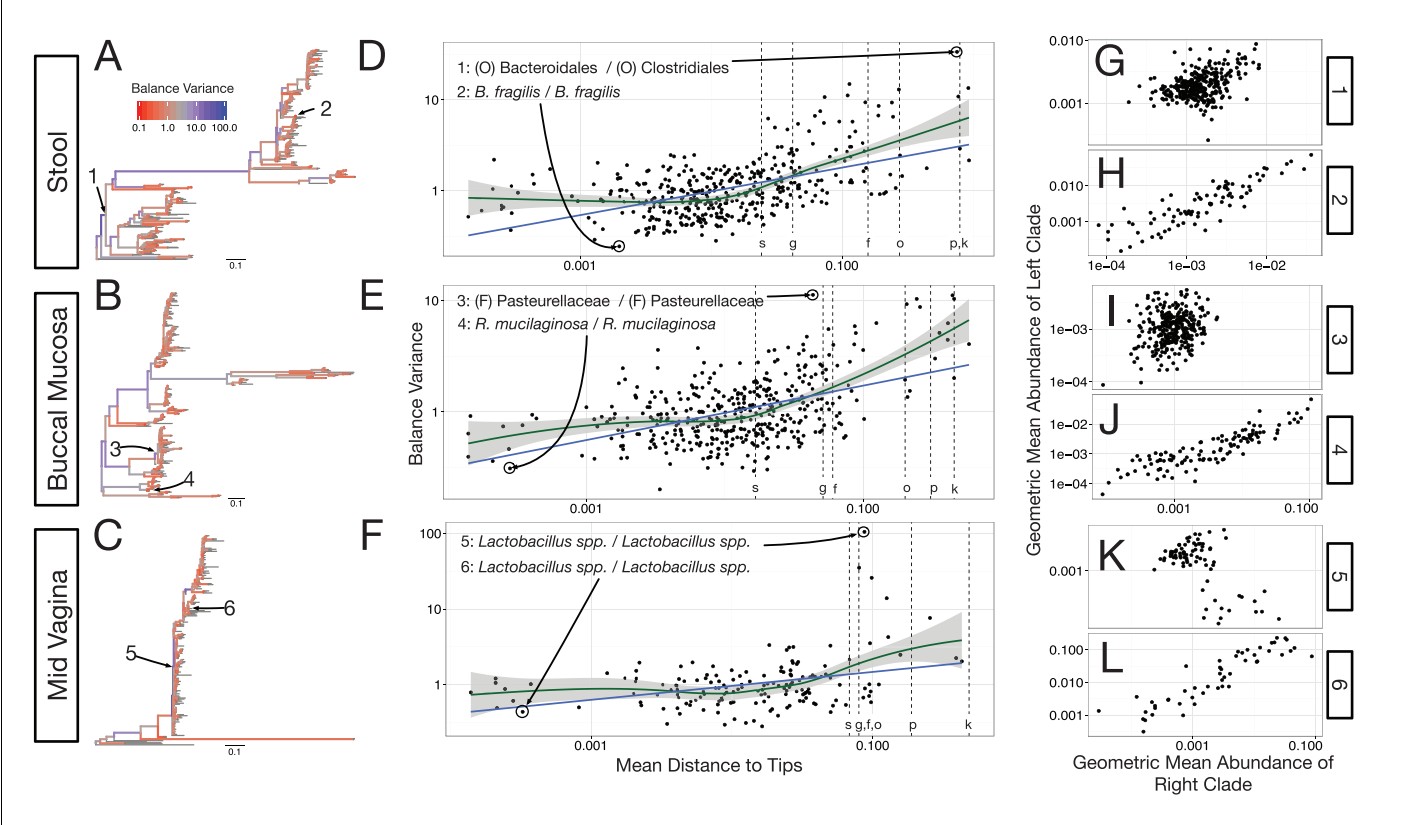

**Figure 4.** Neighboring clades covary less with increasing phylogenetic depth. The variance of balance values captures the degree to which neighboring clades covary, with smaller balance variances representing sister clades that covary more strongly (*Figure 4—figure supplement 1*). (A–C) Balance variances were computed among samples from stool (**A**), buccal mucosa (**B**), and the mid-vagina (**C**). Red branches indicate small balance variance and blue branches indicate high balance variance. Balances 1–6 are individually tracked in panels (**D–L**). (**D–F**) Balance variances within each body site increased linearly with increasing phylogenetic depth on a log-scale (blue line; p<0.01, permutation test with FDR correction; *Methods*). Significant trends are seen across all other body sites (*Figure 4—figure supplements 2* and *3*). Non-parametric LOESS regression (green line and corresponding 95% confidence interval) reveals an inflection point in the relation between phylogenetic depth and balance variance. This inflection point appears below the estimated species level ('s' dotted line; the median depth beyond which balances no longer involve leaves sharing the same species assignment; Materials and methods). (**G–L**) Examples of balances with high and low variance from panels (**A–F**). Low balance variances (**H, J, L**) reflect a linear relationship between the geometric means of sister clades abundances. High balance variances (**G, I, K**) reflect either unlinked or exclusionary dynamics between the geometric means of sister clades abundances.

The following source data, source code and figure supplements are available for figure 3:

**Source code 1.** Source code for *Figure 4* and associated supplements.
**Source data 1.** FDR corrected p-values from permutation tests.
**Figure supplement 1.** Balances with high and low variance.
**Figure supplement 2.** Neighboring clades covary less with increasing phylogenetic depth.
**Figure supplement 3.** The null distribution for *β*.

## Discussion

There exists a symbiosis between our understanding of bacterial evolution and the ecology of host-associated microbial communities (*Matsen, 2015*). Microbiota studies have shown that mammals and bacteria cospeciated over millions of years (*Moeller et al., 2016*; *Ley et al., 2008*), and human

gut microbes have revealed the forces driving horizontal gene transfer between bacteria (*Smillie et al., 2011*). Evolutionary tools and theory have been used to explain how cooperation benefits members of gut microbial communities (*Rakoff-Nahoum et al., 2016*), and raise concerns that rising rates of chronic disease are linked to microbiota disruption (*Blaser and Falkow, 2009*). Here, we aimed to continue building links between microbiota evolution and ecology by designing a data transform that uses phylogenetic models to overcome the challenges associated with compositional data while enabling novel evolutionary analyses.

We found that the resulting PhILR coordinate system, at least with respect to the performance metrics chosen, led to significantly improved performance for a variety of community-level analyses now used in microbiota analysis. While these results add credence to our proposed approach, we underscore that we do not find it essential that PhILR demonstrates superior benchmark performance to motivate its use in microbiota analysis. We believe that the need for compositionally robust tools has already been well established (*Jackson, 1997*; *Friedman and Alm, 2012*; *Aitchison, 1986*; *Lovell et al., 2011*; *Gloor et al., 2016a*; *Li, 2015*; *Tsilimigras and Fodor, 2016*) and intended these benchmarks to showcase the flexibility and utility of working with PhILR transformed data. We also note that for some analyses, a phylogeny-based ILR transform will not outperform an ILR transform built from another sequential binary partition. In fact, in the absence of branch length weights, any random ILR partition would yield equivalent results on our benchmark tasks. Instead, what distinguishes the PhILR transform from other ILR transforms is the interpretability of the transformed coordinates. Balances in PhILR space correspond to speciation events, which can be a source for biological insight.

For example, performing regression on PhILR transformed data enabled us to decompose the distance between bacterial communities onto individual locations on the phylogeny, highlighting balances near the tips of the tree that distinguished human body sites. These balances may reflect functional specialization, as ecological partitioning among recently differentiated bacterial clades could be caused by genetic adaptation to new environments or lifestyles (*Hunt et al., 2008*). Indeed, among oral body sites, we observed consistent site specificity of neighboring bacterial clades within the genera *Actinomyces* (*Figure 3F*) and *Streptococcus* (*Figure 3E*). Species within the *Actinomyces* genera have been previously observed to partition by oral sites (*Aas et al., 2005*; *Mager et al., 2003*). Even more heterogeneity has been observed within the *Streptococcus* genus, where species have been identified that distinguish teeth, plaque, mucosal, tongue, saliva, and other oral sites (*Aas et al., 2005*; *Mager et al., 2003*). This partitioning likely reflects variation in the anatomy and resource availability across regions of the mouth (*Aas et al., 2005*), as well as the kinds of surfaces bacterial strains can adhere to (*Mager et al., 2003*).

We also observed evidence for potential within-genus adaptation to body sites that has not been previously reported. Within the genus *Corynebacterium*, we found ratios of taxa varied among oral plaques and select skin sites (*Figure 3D*). Although the genus is now appreciated as favoring moist skin environments, the roles played by individual Corynebacteria within skin microbiota remain incompletely understood (*Grice and Segre, 2011*). Precisely linking individual *Corynebacterium* species or strains to body sites is beyond the scope of this study due to the limited taxonomic resolution of 16S rRNA datasets (*Janda and Abbott, 2007*; *Větrovský and Baldrian, 2013*). Nevertheless, we believe the PhILR coordinate system may be used in the future to identify groups of related bacterial taxa that have undergone recent functional adaptation.

Another example of how the PhILR transform can provide biological insights arose in our analysis of how human microbiota variance decomposes along individual balances. We observed that balances between more phylogenetically related clades were significantly more likely to covary than expected by chance. This pattern could reflect evolutionary and ecological forces structuring microbial communities in the human body. Related bacterial taxa have been hypothesized to have similar lifestyle characteristics (*Martiny et al., 2015*; *Zaneveld et al., 2010*), and may thus covary in human body sites that favor their shared traits (*Levy and Borenstein, 2013*; *Faust et al., 2012*). An alternative explanation for the balance variation patterns we observed is that sequencing errors and read clustering artifacts are likely to produce OTUs (Operational Taxonomic Units) with similar reference sequences and distributions across samples. While we cannot conclusively rule out this alternative hypothesis, we note that it would not explain why signal for taxa co-variation is weakest for balances at higher taxonomic levels and appears to plateau for balances near or below the species level. A biological explanation for the plateauing signal would be that lifestyle characteristics enabling

bacteria to persist in human body sites are conserved among strains roughly corresponding to the same species. Follow-up studies are needed to more conclusively understand how balance variance patterns across the phylogenies can be interpreted from an evolutionary standpoint.

Though the methods presented here provide a coherent geometric framework for performing microbiota analysis free from compositional artifacts, future refinements are possible. Specifically, we highlight issues relating to our choice of weights, the handling of zero values, and information loss during count normalization. Both the taxa weights and the branch lengths weights we introduce here may be viewed as preliminary heuristics; future work will likely yield additional weighting schemes, as well as knowledge for when a given weighting scheme should be matched to an analysis task. In the case of supervised machine learning, weighting selection could be optimized as part of the training process. Additionally, if it is important that the transformed data has meaningful numerical coordinates, such that one desires to interpret the exact numerical value of a given balance in a sample, we suggest that neither branch length weights nor taxa weights be used as these weights can complicate this type of interpretation. Concerning our handling of zero values, this model design choice confronts an outstanding challenge for microbiota and compositional data analysis (*Tsilimigras and Fodor, 2016*; *Martın-Fernandez et al., 2011*). Part of this challenge's difficulty is whether a zero value represents a value below the detection limit (rounded zero) or a truly absent taxon (essential zero). Here, we employ zero replacement, which implies an assumption that all zero values represent rounded zeros. New mixture models that explicitly allow for both essential and rounded zeros (*Bear and Billheimer, 2016*), as well as more advanced methods of zero replacement (*Martın-Fernandez et al., 2011*; *Martin-Fernandez et al., 2015*), may enable us to handle zeros in a more sophisticated manner. Lastly, in regards to informational loss caused by normalization, it is known that the number of counts measured for a given taxon influences the precision with which we may estimate its relative abundance in a sample (*Gloor et al., 2016a*; *Good, 1956*; *McMurdie and Holmes, 2014*). While our taxa weights are intended to address this idea, a fully probabilistic model of counts would likely provide more accurate error bounds for inference. We believe it would be possible to build such a model in a Bayesian framework by viewing the observed counts as multinomial draws from a point in the PhILR transformed space, as has been done for other log-ratio based spaces (*Billheimer et al., 2001*).

Beyond refining the PhILR transform itself, future effort may also be directed towards interpreting the transform's results at the single taxon level. Microbiota studies frequently focus on individual taxa for tasks such as identifying specific bacteria that are causal or biomarkers of disease. Log-ratio approaches can provide a compositionally robust approach to identifying biomarkers based on changes in the relative abundance of individual taxa. Due to the one-to-one correspondence between CLR coordinates and individual taxa, the CLR transform has been used previously to build compositionally robust models in terms of individual taxa (*Mandal et al., 2015*; *Kurtz et al., 2015*; *Fernandes et al., 2014*). However, CLR transformed data suffer from the drawback of a singular covariance matrix, which can make the development of new models based on the CLR transform difficult (*Pawlowsky-Glahn et al., 2015*). ILR transformed data do not suffer this drawback (*Pawlowsky-Glahn et al., 2015*) and moreover, can be analyzed at the single taxon level. To do so, the inverse ILR transform can be applied to model results generated in an ILR coordinate system, yielding analyses in terms of changes in the relative abundance of individual taxa (*Pawlowsky-Glahn et al., 2015*). The use of the inverse ILR transform in this manner is well established (*Pawlowsky-Glahn et al., 2015*; *van den Boogaart and Tolosana-Delgado, 2013*; *Pawlowsky-Glahn and Buccianti, 2011*) and the inverse transform is provided in the *Methods* (*Egozcue and Pawlowsky-Glahn, 2016*).

Despite these avenues for improvement, modification, or extension we believe the PhILR transform already enables existing statistical methods to be applied to metagenomic datasets, free from compositional artifacts and framed according to an evolutionary perspective. We foresee the PhILR transform being used as a default transformation prior to many microbiota analyses, particularly if a phylogenetic perspective is desired. For example, the PhILR transform could be used in lieu of the conventional log transform, which is often the default choice in microbiota analysis but is not robust to compositional effects. Substituting PhILR into existing bioinformatics pipelines should often be seamless and we emphasize that all statistical tools applied to PhILR transformed data in this study were used 'off-the-shelf' and without modification. Importantly, such a substitution contrasts with the alternative approach for accounting for compositional microbiota data, which is to modify

existing statistical techniques. Such modification is often challenging because many statistics were derived assuming an unconstrained space with an orthonormal basis, not a constrained and over-determined compositional space. Therefore, while select techniques have already been adapted (e. g. distance measures that incorporate phylogenetic information (*Lozupone and Knight, 2005*) and feature selection methods that handle compositional input (*Chen and Li, 2013*; *Lin et al., 2014*)), it is likely that certain statistical goals, such as non-linear community forecasting or control system modeling, may prove too complex for adapting to the compositional nature of microbiota datasets. Finally, beyond microbiota surveys, we also recognize that compositional metagenomics datasets are generated when studying the ecology of viral communities (*Culley et al., 2006*) or clonal population structure in cancer (*Britanova et al., 2014*; *Yuan et al., 2015*; *Roth et al., 2014*). We expect the PhILR transform to aid other arenas of biological research where variables are measured by relative abundance and related by an evolutionary tree.

## Materials and methods

### The ILR transform

A typical microbiome sample consists of measured counts $c_j$ for taxa $j \in \{1, \ldots, D\}$. A standard operation is to take count data and transform it to relative abundances. This operation is referred to as closure in compositional data analysis (*Aitchison, 1986*) and is given by

$$\boldsymbol{x} = C[(c_1, \cdots, c_D)] = \left( \frac{c_1}{\sum_j c_j}, \ldots, \frac{c_D}{\sum_j c_j} \right)$$

where $\boldsymbol{x}$ represents a vector of relative abundances for the $D$ taxa in the sample. We can represent a binary phylogenetic tree of the $D$ taxa using a sign matrix $\Theta$ as introduced by Pawlowsky-Glahn and Egozcue (*Pawlowsky-Glahn and Egozcue, 2011*) and shown in *Figure 5*. Each row of the sign matrix indexes an internal node $i$ of the tree and each column indexes a tip of the tree. A given element in the sign matrix is $\pm 1$ depending on which of the two clades descending from $i$ that tip is a part of and 0 if that tip is not a descendent of $i$. The assignment of $+1$ versus $-1$ determines which clade is represented in the numerator versus the denominator of the corresponding log-ratio (as described below). Exchanging this assignment for a given balance switches which clade is represented in the numerator versus the denominator of the log-ratio. Following Egozcue and Pawlowsky-Glahn (*Egozcue and Pawlowsky-Glahn, 2016*), we represent the ILR coordinate (balance) associated with node $i$ in terms of the shifted composition $\boldsymbol{y} = \boldsymbol{x}/\boldsymbol{p} = (x_1/p_1, \ldots, x_D/p_D)$ as

$$y_i^* = \sqrt{\frac{n_i^+ n_i^-}{n_i^+ + n_i^-}} \log \frac{g_p(\boldsymbol{y}_i^+)}{g_p(\boldsymbol{y}_i^-)} \; . \tag{1}$$

Here, $g_p(\boldsymbol{y}_i^+)$ and $g_p(\boldsymbol{y}_i^-)$ represents the weighted geometric mean of the components of $y$ that represent tips in the $+1$ or $-1$ clade descendant from node $i$ respectively. This weighted geometric mean is given by

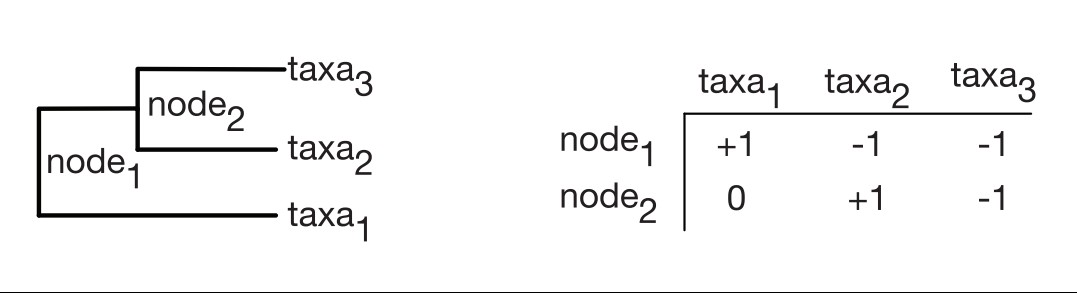

**Figure 5.** Sign matrix representation of a phylogenetic tree. A binary tree (**Left**) can be represented by a sign matrix (**Right**) denoted $\Theta$.

$$g_p(\boldsymbol{y}_i^{\pm}) = \exp\left(\frac{\sum_{(\theta_{ij}=\pm 1)} p_j \log y_j}{\sum_{(\theta_{ij}=\pm 1)} p_j}\right) \tag{2}$$

where $p_j$ is the weight assigned to taxa $j$. The term $\sqrt{n_i^+ n_i^- / n_i^+ + n_i^-}$ in *equation 1* is the scaling term that ensures that the ILR basis element has unit length and the terms $n_i^{\pm}$ are given by

$$n_i^{\pm} = \sum_{\theta_{ij}=\pm 1} p_j. \tag{3}$$

Note that when $\boldsymbol{p} = (1,\ldots,1)$, $\boldsymbol{y} = \boldsymbol{x}$, *Equation 1* represent the ILR transform as originally published (*Egozcue et al., 2003*), *Equation 2* represents the standard formula for geometric mean of a vector $\boldsymbol{y}$, and *equation 3* represents the number of tips that descend from the $+1$ or $-1$ clade descendant from node $i$. However, when $\boldsymbol{p} \neq (1,\ldots,1)$, these three equations represent a more generalized form of the ILR transform that allows weights to be assigned to taxa in the transformed space (*Egozcue and Pawlowsky-Glahn, 2016*).

Following Egozcue and Pawlowsky-Glahn (*Egozcue and Pawlowsky-Glahn, 2016*), we also note that the form of the generalized ILR (which we will denote $\mathrm{ilr_p}$) transform can be rewritten in terms of a generalized CLR transform (which we will denote $\mathrm{clr_p}$). This formulation in terms of the generalized CLR transform can be more efficient to compute and allows the inverse of the transform to be easily described. We can define the generalized CLR transform as

$$\mathrm{clr_p}(\boldsymbol{y}) = \left(\log\frac{y_1}{g_p(\boldsymbol{y})}, \cdots, \log\frac{y_D}{g_p(\boldsymbol{y})}\right).$$

The generalized ILR transform can then be written as

$$\boldsymbol{y}^* = \mathrm{ilr_p}(\boldsymbol{y}) = \mathrm{clr_p}(\boldsymbol{y})\,\mathrm{diag}(\boldsymbol{p})\Psi^T$$

with the *ij*th element of the matrix $\Psi$ given by

$$\psi_{ij} = \begin{cases} +\frac{1}{n_i^+}\sqrt{\frac{n_i^+ n_i^-}{n_i^+ + n_i^-}} & \text{if } \theta_{ij} = +1 \\ -\frac{1}{n_i^-}\sqrt{\frac{n_i^+ n_i^-}{n_i^+ + n_i^-}} & \text{if } \theta_{ij} = -1 \\ 0 & \text{if } \theta_{ij} = 0\,. \end{cases}$$

With these components defined the inverse of generalized ILR transform can be written as $\mathcal{C}[\boldsymbol{y}] = \mathrm{ilr_p}^{-1}(\boldsymbol{y}^*) = \mathcal{C}[\exp(\boldsymbol{y}^*\Psi)]$ and $x = \mathcal{C}\left[\mathrm{ilr_p}^{-1}(\boldsymbol{y}^*)\boldsymbol{p}\right]$.

## Soft thresholding through weighting taxa

We make use of this generalized ILR transform to down weight the influence of taxa with many zero and near-zero counts since these are less reliable and therefore more variable (*Good, 1956*). Our choice of taxa weights is a heuristic that combines two terms multiplicatively: a measure of the central tendency of counts, such as the mean or median of the raw counts for a taxon across the $N$ samples in a dataset; and, the norm of the vector of relative abundances of a taxon across the $N$ samples in a dataset. We add this vector norm term to weight taxa by their site-specificity. Preliminary studies showed that the geometric mean of the counts (with a pseudocount added to avoid skew from zero values) outperformed both the arithmetic mean and median as a measure of central tendency for the counts (data not shown). Additionally, while both the Euclidean norm and the Aitchison norm improved preliminary benchmark performance compared to using the geometric mean alone, in one case (classification using support vector machine on the global patterns dataset), the Euclidean norm greatly outperformed the Aitchison norm (*Supplementary file 1*). Therefore, our chosen taxa weighting scheme uses the geometric mean times the Euclidean norm:

$$p_j = \sqrt[N]{(c_{j1}+1)\cdot\ldots\cdot(c_{jN}+1)} \cdot \|x_j\|.$$

Note that we add the subscript $j$ to the right-hand side of the above equation to emphasize that

this is calculated with respect to a single taxon across the $N$ samples in a dataset. As intended, this scheme tended to assign smaller weights to taxa in our benchmarks with more zero and near-zero counts (*Figure 2—figure supplement 1*). Despite their heuristic nature, we found that our chosen weights provide performance improvements over alternative weights (or the lack thereof) as measured by our benchmark tasks (*Supplementary file 1*).

Our taxa weighting scheme supplements the use of pseudo-counts and represents a soft-threshold on low abundance taxa. More generally, these taxa weights represent a form of prior information regarding the importance of each taxon. We note that if prior biological information suggests allowing specific taxa to influence the PhILR transform more (or less) strongly, such a weighting could be achieved for taxon $j$ by increasing (or decreasing) $p_j$.

## Incorporating branch lengths

Beyond utilizing the connectivity of the phylogenetic tree to dictate the partitioning scheme for ILR balances, branch length information can be embedded into the transformed space by linearly scaling ILR balances ($y_i^*$) by the distance between neighboring clades. We call this scaling by phylogenetic distance 'branch length weighting'. Specifically, for each coordinate $y_i^*$, corresponding to node $i$ we use the transform

$$y_i^{*,blw} = y_i^* \cdot f\left(d_i^+, d_i^-\right)$$

where $d_i^{\pm}$ represent the branch lengths of the two direct children of node $i$. When $f\left(d_i^+, d_i^-\right) = 1$, the coordinates are not weighted by branch lengths. The form of this transform was chosen so that the weights $d_i^{\pm}$, only influence the corresponding coordinate ($y_i^{*,blw}$).

We also investigated the effect of using $f\left(d_i^+, d_i^-\right) = 1$, $f\left(d_i^+, d_i^-\right) = d_i^+ + d_i^-$, and based on the results of *Chen et al. (2012)*, $f\left(d_i^+, d_i^-\right) = \sqrt{d_i^+ + d_i^-}$ on benchmark performance. When coupled with the taxa weights specified above, the square root of the summed distances had the highest rank in 9 of the 12 supervised classification tasks and 2 of the three distance based tasks (*Supplementary file 1*). Based on these results, except for our analysis of balance variance versus phylogenetic depth (see below), the square root of the summed distances was used throughout our analyses.

## Implementation

The PhILR transform, as well as the incorporation of branch length and taxa weightings has been implemented in the R programing language as the package *philr* available at https://bioconductor.org/packages/philr/.

## Datasets and preprocessing

All data preprocessing was done in the R programming language using the *phyloseq* package for analysis of microbiome census data (*McMurdie and Holmes, 2013*) as well as the *ape* (*Paradis et al., 2004*) and *phangorn* (*Schliep, 2011*) packages for analysis of phylogenetic trees.

### Data acquisition

We chose to use previously published OTU tables, taxonomic classifications, and phylogenies as the starting point for our analyses. The Human Microbiome Project (HMP) dataset was obtained from the QIIME Community Profiling Pipeline applied to high-quality reads from the v3-5 region, available at http://hmpdacc.org/HMQCP/. The Global Patterns dataset was originally published in Caporaso, et al. (*Caporaso et al., 2011*) and is provided with the *phyloseq* R package (*McMurdie and Holmes, 2013*). The Costello Skin Sites dataset (CSS) is a subset of the dataset collected by Costello et al. (*Costello et al., 2009*) featuring only the samples from skin sites. This skin subset was introduced as a benchmark for supervised machine learning by Knights et al. (*Knights et al., 2011*) and can be obtained from http://www.knightslab.org/data.

### OTU table preprocessing

To accord with general practice, we performed a minimal level of OTU table filtering for all datasets used in benchmarks and analyses. Due to differences in sequencing depth, sequencing methodology, and the number and diversity of samples between datasets, filtering thresholds were set

independently for each dataset. For the HMP dataset, we initially removed samples with fewer than 1000 counts to mimic prior analyses (*Human Microbiome Project Consortium, 2012*). We additionally removed OTUs that were not seen with more than three counts in at least 1% of samples. Preprocessing of the Global Patterns OTU table followed the methods outlined in McMurdie and Holmes (*McMurdie and Holmes, 2013*). Specifically, OTUs that were not seen with more than three counts in at least 20% of samples were removed, the sequencing depth of each sample was standardized to the abundance of the median sampling depth, and finally OTUs with a coefficient of variation ≤3.0 were removed. The CSS dataset had lower sequencing depth than the other two datasets; we chose to filter OTUs that were not seen with greater than 10 counts across the skin samples. The PhILR transform, and more generally our benchmarking results in *Figure 2b and C*, were robust to varying our filtering strategies (*Supplementary file 2*).

### Preprocessing phylogenies
For each dataset, the phylogeny was pruned to include only those taxa remaining after OTU table preprocessing. Except for the Global Patterns dataset, which was already rooted, we chose to root phylogenies by manually specifying an outgroup. For the HMP dataset the phylum Euryarchaeota was chosen as an outgroup. For the CSS dataset, the tree was rooted with OTU 12871 (from phylum Plantomycetes) as the outgroup. For all three phylogenies, any multichotomies were resolved with the function multi2di from the *ape* package which replaces multichotomies with a series of dichotomies with one (or several) branch(es) of length zero.

### Zero replacement and normalization
A pseudocount of 1 was added prior to PhILR transformation to avoid taking log-ratios with zero counts. We found that our benchmarking results were robust to changing the value of this pseudocount from 1 to 2, 3, or 10 (*Supplementary file 1*).

### Grouping sampling sites
To simplify subsequent analyses, HMP samples from the left and right retroauricular crease and samples from the left and right antecubital fossa were grouped together, respectively, as preliminary PERMANOVA analysis suggested that these sites were indistinguishable (data not shown).

## Benchmarking
### Distance/dissimilarity based analysis
Distance between samples in PhILR transformed space was calculated using Euclidean distance. All other distance measures were calculated using *phyloseq* on the preprocessed data without adding a pseudocount. Principle coordinate analysis was performed for visualization using *phyloseq*. PERMANOVA was performed using the function *adonis* from the R package *vegan* (v2.3.4). The $R^2$ value from the fitted model was taken as a performance metric. Standard errors were calculated using bootstrap resampling with 100 samples each. Differences between the performance of Euclidean distance in PhILR transformed space and that of each other distance or dissimilarity measure on a given task was tested using two-sided t-tests and multiple hypothesis testing was accounted for using FDR correction.

### Supervised classification
The performance, as measured by classification accuracy, of PhILR transformed data was compared against data preprocessed using one of two standard strategies for normalizing sequencing depth: the preprocessed data was transformed to relative abundances (*e.g.*, each sample was normalized to a constant sum of 1; *raw*); or, a pseudocount of 1 was added, the data was transformed to relative abundances, and finally the relative abundances were log-transformed (*log*).

All supervised learning was implemented in Python using the following libraries: *Scikit-learn* (v0.17.1), *numpy* (v1.11.0) and *pandas* (v0.17.1). Four classifiers were used: penalized logistic regression, support vector classification with RBF kernel, random forest classification, and k-nearest-neighbors classification. Each classification task was evaluated using the mean and variance of the test accuracy over 10 randomized test/train (30/70) splits which preserved the percentage of samples from each class at each split. For each classifier, for each split, the following parameters were set

using cross-validation on the training set. Logistic regression and Support Vector Classification: the 'C' parameter was allowed to vary between $10^{-3}$ to $10^3$ and multi-class classification was handled with a one-vs-all loss. In addition, for logistic regression the penalty was allowed to be either $l_1$ or $l_2$. K-nearest-neighbors classification: the 'weights' argument was set to 'distance'. Random forest classification: each forest contained 30 trees and the 'max_features' argument was allowed to vary between 0.1 and 1. All other parameters were set to default values. Due to the small size of the Global Patterns dataset, the supervised classification task was simplified to distinguishing human vs. non-human samples. Differences between each method's accuracy in each task was tested using two-sided t-tests and multiple hypothesis testing was accounted for using FDR correction.

## Identifying balances that distinguish sites

To identify a sparse set of balances that distinguish sampling sites while accounting for the dependencies between nested balances, we fit a multinomial regression model with a grouped l₁ penalty using the R package *glmnet* (*v2.0.5*). The penalization term lambda was set by visually inspecting model outputs for clear body site separation (lambda = 0.1198). This resulted in 35 balances with non-zero regression coefficients. Phylogenetic tree visualization was done using the R package *ggtree* (*Yu et al., 2017*).

## Variance and depth

To reduce the likelihood that our analysis of balance variance and phylogenetic depth was affected by statistical artifact, we modified our PhILR transform in several ways. First, we omitted branch length weights (*i.e.*, we set $f(d_i^+, d_i^-) = 1$) as these may vary non-randomly as a function of phylogenetic depth. Second, we also anticipated that any zero replacement method would likely lead to lower variance measurements, which could have greater effects on balances closer to the tips of the tree. We therefore omitted taxa weights and zero replacement; we instead used stricter hard filtering thresholds and calculated balance values based on non-zero counts. In practice, we used the following filtering thresholds for each body site, taxa present in less than 20% of samples from that site were excluded and subsequently samples that had less than 50 total counts were excluded. To calculate balance values based on non-zero counts we retained balances that met the following criteria: the term $g_p(\boldsymbol{y}_i^+)/g_p(\boldsymbol{y}_i^-)$ had non-zero counts from some taxa within the subcomposition $\boldsymbol{y}_i^+$ (formed by the taxa that descend from the +1 clade of node $i$) and some other taxa within the subcomposition $\boldsymbol{y}_i^-$ (formed by the taxa that descend from the −1 clade of node $i$) in at least 40 samples from that body site. We believe these two modifications to PhILR resulted in a more conservative analysis of balance variance versus phylogenetic depth but are likely not optimal in other situations.

To investigate the overall relationship between balance variance and phylogenetic depth we used linear regression. A balance's depth in the tree was calculated as its mean phylogenetic distance to its descendant tips $(d)$. For a given body site the following model was fit:

$$\log var(y^*) = \beta \log d + \alpha$$

where $d$ represents mean distance from a balance to its descendant tips. We then set out to test the null hypothesis that $\beta = 0$, or that the variance of the log-ratio between two clades was invariant to the distance of the two clades from their most recent common ancestor. For each site, a null distribution for $\beta$ was constructed by permutations of the tip labels of the phylogenetic tree. For each permutation of the labels, the resultant tree was used to transform the data and $\beta$ was estimated. We chose this permutation scheme to ensure that the increasing variance we saw with increasing proximity of a balance to the root was not because deeper balances had more descendant tips, an artifact of variance scaling with mean abundance, or due to bias introduced due to our handling of zeros. Furthermore, for each body site, we found the null distribution for $\beta$ was symmetric about $\beta = 0$ which further supports that balance variance depends on phylogenetic depth through a biological mechanism and not through a statistical artifact (*Figure 4—figure supplement 3*). Two tailed p-values were calculated for $\beta$ based on 20000 samples from each site's respective null distribution. FDR correction was applied to account for multiple hypothesis testing between body sites.

To visualize local trends in the relationship between balance variance and phylogenetic depth, a LOESS regression was fit independently for each body site. This was done using the function *geom_smooth* from the R package ggplot2 (v2.1.0) with default parameters.

The data and code needed to reproduce our analysis of balance variance versus phylogenetic depth is provided in *Figure 4—source data 1* and *Figure 4—source code 1* respectively.

### Integrating taxonomic information

Taxonomy was assigned to OTUs in the HMP dataset using the *assign_taxonomy.py* script from *Qiime* (v1.9.1) to call *uclust* (v1.2.22) with default parameters using the representative OTU sequences obtained as described above. Taxonomic identifiers were assigned to the two descendant clades of a given balance separately using a simple voting scheme and combined into a single name for that balance. The voting scheme occurs as follows: (1) for a given clade, the entire taxonomy table was subset to only contain the OTUs that were present in that clade (2) starting at the finest taxonomic rank the subset taxonomy table was checked to see if any species identifier represented ≥95% of the table entries at that taxonomic rank, if so that identifier was taken as the taxonomic label for the clade (3) if no consensus identifier was found, the table was checked at the next most-specific taxonomic rank.

Median phylogenetic depths for each taxonomic rank were estimated by first decorating a phylogenetic tree with taxonomy information using *tax2tree* (v1.0) (*McDonald et al., 2012*). For a given taxonomic rank the mean distance to tips was calculated for each internal node possessing a label that ended in that rank. The median of these distances was used to display an estimate of the phylogenetic depth of that given rank. This calculation of median phylogenetic depth of different taxonomic ranks was done separately for each body site.

The data and code needed to reproduce the taxonomic assignment and estimation of median phylogenetic depths for each taxonomic rank is included in *Figure 4—source data 1* and *Figure 4—source code 1* respectively.

## Acknowledgements

We thank Rachel Silverman, Aspen Reese, Firas Midani, Heather Durand, Jesse Shapiro, Jonathan Friedman, Simon Levin, and Susan Holmes for their helpful comments, Dan Knights for providing us with the CSS dataset, as well as Klaus Schliep and Liam Revell for their insight into manipulation of phylogenetic trees in the R programming language. JS was supported in part by the Duke University Medical Scientist Training Program (GM007171).

## Additional information

### Funding

| Funder | Grant reference number | Author |
| --- | --- | --- |
| National Science Foundation | IIS 1546331 | Sayan Mukherjee |
| National Science Foundation | IIS 1418261 | Sayan Mukherjee |
| Global Probiotics Council | Young Investigator Grant for Probiotics Research | Lawrence A David |
| Searle Scholars Program | 15-SSP-184 Research Agreement | Lawrence A David |
| Alfred P. Sloan Foundation | BR2014-003 | Lawrence A David |
| Duke University | Medical Scientist Training Program GM007171 | Justin D Silverman |

The funders had no role in study design, data collection and interpretation, or the decision to submit the work for publication.

### Author contributions

JDS, Conceptualization, Software, Formal analysis, Investigation, Visualization, Methodology, Writing—original draft, Writing—review and editing; ADW, Conceptualization, Investigation, Methodology, Writing—original draft, Writing—review and editing; SM, Formal analysis, Supervision, Funding acquisition, Methodology, Writing—original draft, Writing—review and editing; LAD,

Conceptualization, Resources, Formal analysis, Supervision, Funding acquisition, Investigation, Writing—original draft, Project administration, Writing—review and editing

### Author ORCIDs
Justin D Silverman, http://orcid.org/0000-0002-3063-2098
Lawrence A David, http://orcid.org/0000-0002-3570-4767

## Additional files

### Supplementary files
• Supplementary file 1. Extended benchmarking results including the effects of changing pseudo-count and different choices for taxa and branch length weights. This file also contains summary information for the benchmark datasets after preprocessing.

• Supplementary file 2. Extended benchmarking results regarding the sensitivity of methods to different OTU filtering schemes.

### Major datasets
The following previously published datasets were used:

| Author(s) | Year | Dataset title | Dataset URL | Database, license, and accessibility information |
|---|---|---|---|---|
| Human Microbiome Project Consortium | 2010 | Human Microbiome Project | http://hmpdacc.org/HMQCP/ | Publicly available at HMPDACC (v35 download of files 6, 9, and 10). |
| Costello EK, Lauber CL, Hamady M, Fierer N, Gordon JI, Knight R | 2009 | Costello Skin Sites | http://www.knightslab.org/data | Publicly available as part of the FEMS Benchmark dataset (2011) provided by Dan Knights |
| Caporaso GJ, Lauber CL, Walters WA, Berg-Lyons D, Lozupone CA, Turnbaugh PJ, Fierer N, Knight R | 2011 | Global Patterns | https://bioconductor.org/packages/release/bioc/html/phyloseq.html | Publicly available and provided as part of the phyloseq R package as 'GlobalPatterns' |

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
