## [Decision Letter]

Thank you for submitting your article "A phylogenetic transform enhances analysis of compositional microbiota data" for consideration by *eLife*. Your article has been favorably evaluated by Wendy Garrett (Senior Editor) and three reviewers, one of whom, Anthony Fodor, is a member of our Board of Reviewing Editors. The following individual involved in review of your submission has agreed to reveal their identity: Juan José Egozcue (Reviewer #2). All three reviewers found considerable merit in the manuscript praising the originality of the idea, the clarity of the presentation and the effectiveness of the visualizations.

For your reference, I have attached the three original reviews below. Per the *eLife* policy, you are not required to respond to each individual point raised by each reviewer. Below you will find a summary of the reviews reflecting the consensus of the reviewers of required revisions.

Summary:

It has long been recognized that compositional data is best analyzed in ratio space. For 16S rRNA sequence datasets and other genomic datasets, this leads to the rather difficult question of what ratio to take. In this interesting manuscript, Silverman et al. suggest taking ratios on a bifurcating phylogenetic tree. In their approach, the dependent variable for each node of the tree is a "balance", which is a normalized ratio of all the nodes beneath each node (with the ratio of all the left descendant nodes over all the right descendant nodes). The paper explores the consequences of this normalization showing reasonable performance when used for classification or machine learning. The authors provide an implementation in R and this approach may be of utility for analysts looking for a novel way to analyze microbial data in ratio space.

Essential revisions:

1) There was consensus among the reviewers that the normalization scheme seemed a bit ad-hoc and poorly justified. We suggest that it is made clear to the readers that the choice of weights used in the manuscript, both on the taxa and on the tips, are preliminary choices and further work will be required to ultimately determine the best weighting scheme. In addition, some further clarification on how sparsity was treated would strengthen the manuscript.

2) None of the reviewers were entirely satisfied with the demonstration that the proposed transformation was superior to other transformations. It is unclear what it means, for example, that the PhILR technique sometimes produces a stronger clustering signal and sometimes a weaker clustering signal than weighted UNIFRAC (Figure 2). There was some discussion among the reviewers about whether a comparison to random binary partitions would be appropriate since this could reflect your null model (that phylogenetic binary partition is superior). Reviewer #2 was not enthusiastic about this idea and argued that a change on a tree does not alter (Aitchison) distances between data points. Therefore, results would likely not change no matter which tree has been selected for defining ILR coordinates.

In the end, the consensus of the reviewers was that exploring such null models might be of interest, but would not be appropriate as a requirement for revision. Nonetheless, reviewer #1 would have been interested to see if machine learning algorithms in fact performed better on the phylogenetic partition in comparison to random partitions. We leave it up to the authors as to whether they feel such an inclusion would improve their paper or be worth the effort. However, all reviewers felt that paper would be improved by more discussion of how to tell when one normalization procedure is superior to another. It seemed to reviewer #1 that it might be adequate to say that the proposed transformation appears to be doing no harm relative to weighted Unifrac but was providing some compositional safeguards. Reviewer #2 argued that the main benefit of the proposed procedure was in interpretability in yielding a phylogenetically aware transformation that could be utilized for test-statistics.

3) There was general consensus among the reviewers that Figure 4 was unconvincing in that technical (non-biological) features of the algorithm could explain the results more easily than exclusionary dynamics. The authors could likely remove Figure 4 without damaging the paper, but if they choose to keep it, they should respond to the concerns below from reviewers #1 and #3 as both reviewers felt that simple lack of association between taxa and stochastic variation in sampling could explain many of the observations in Figure 4.

4) There was consensus among the reviewers that the authors could provide more guidance to the non-specialists about when this algorithm should be used. As reviewer #3 put it, when and why should a reader choose the PhILR approach and when should it be avoided.

5) The authors should be explicit about what filtering and other pre-processing steps were used and whether those choices substantially impacted the performance of their algorithm.

*Reviewer #1:*

It has long been recognized that compositional data is best analyzed in ratio space. For 16S rRNA sequence datasets and other genomic datasets, this leads to the rather difficult question of what ratio to take. In this interesting manuscript, Silverman et al. suggest taking ratios on a bifurcating phylogenetic tree. In their approach, the dependent variable for each node of the tree is a "balance", which is a normalized ratio of all the nodes beneath each node (with the ratio of all the left descendant nodes over all the right descendant nodes). This is a clever idea and the authors make a reasonably compelling case that this is a useful contribution. While the evidence from the machine learning and clustering portions of the paper provide only modest evidence that this transformation is substantially better than other transformations, the transformation clearly is not making things much worse. As such, the proposed method will be of interest to researchers interested in how best to approach the challenging problem of compositional analyses of microbial datasets.

Suggestions for improvement follows:

1) The mathematical description of the algorithm could be improved in terms of its clarity. For example, it is not always clear how the pluses and minuses in the second equation of the subsection “The ILR Transform” are defined. It is not made immediately clear that the leftmost term (sqrt(ni+ * ni- / (ni+ + ni-)) in that equation is a correction for branch length. For the equation in the subsection “Addressing sparsity through weighting taxa”, please use parentheses to emphasize that the formula is sqrt(sum(xj)) * ((sum(cj 1…N)) ^ 1/N and not sqrt(sum(xj))^N * sqrt(sum(cj 1…N))

2) A limitation of working in ratio space in 16S data is what to do with all of the zeros in the spreadsheet (the "sparse data" problem). There is nothing inherent to the balancing algorithm that can deal with this problem (the sum in the denominator of the balance obviously can't be zero). The authors therefore require some kind of pseudo-count scheme and they have settled on one (given in the equation in subsection “Addressing sparsity through weighting taxa”) which is a function of the number of samples. If I am reading their formula correctly, the weight given to a taxa with 1 read in one sample and zero in all others would be very close to the sqrt(number of samples). The authors might consider giving this as an example and explicitly noting that their method has some similarity to adding the sqrt(number of samples) as a pseudo-count.

3) I struggled a bit with the variance analysis which concludes the Results sections. In the second paragraph of the subsection 2Balance variance and microbiota assembly” it is argued that "When the variance of a balance between two clades approaches zero, the mean abundance of taxa in each of the two clades will be linearly related and thus exhibit shared dynamics across microbial habitats". While this is true (and must be true mathematically as when the variance approaches zero, the ratio of the two normalized abundances of the clades is constant, which means they will become perfectly linearly related with a positive slope), it's not clear to me that a high variance necessarily means "exclusionary dynamics across samples". I don't see how a high variance in this number indicated anything about how direct or indirect the interactions are within the tree. There are lots of reasons two clades could lack perfect linear correlation with a positive slope from direct competition to being completely unrelated biologically with abundances just shifting due to stochastic variation. Also, wouldn't variance have to be lower closer to the tips of the phylogenetic tree as the number of sequences is lower, so the pseudo-count has more of an effect depressing overall variance? Doesn't that explain Figure 4? And if I am reading these graphs correctly, below the species level there are not enough datapoints to support modeling via localized regression one way or another. Although not central to the arguments of the paper, I found Figure 4 to be less convincing than other sections of the paper.

*Reviewer #2:*

The manuscript introduces two new powerful ideas. They are: (a)construction of the sequential binary partition used to define

ILR coordinates on the base of phylogenetic trees; (b) use therecently published balances in weighted compositions to account forphylogenetic distances between OTU's or taxa. Both ideas aredeveloped and applied successfully to benchmark data sets.

For the originality of main ideas applied to microbiome analysis,the manuscript deserves publication.

*Reviewer #3:*

The manuscript is, in general, written in a clear and logical manner with a minimum of technical jargon. Figure 1, Figure 3 and Figure 4 are outstanding examples of data visualization and demonstrate the utility of this approach over others extant in the field. Figure 5 seems an afterthought and could be combined with Figure 1, or made supplementary. The binary portion could also be completely described in the Methods without a figure.

Figure 2, and the associated descriptive text are somewhat underwhelming demonstrations of the utility of the approach, which, in theory, should clearly outperform all other methods if the compositional data approach is necessary. The use of box-plots probably does not aid the visual appeal.

That the ILR based distance can seriously underperform the weighted unifrac metric is puzzling and cause for concern, since PhILR becomes another metric to be 'mined' to achieve the desired outcome. Does the PhILR partitioning scheme outperform a simple random binary partition? this is crucial to know.

There needs to be serious guidance given to the reader as to when and why to choose the PhILR approach and when not to. The same can be said for the classification methods shown, where the PhILR generally (but not always) outperforms relative abundance, but not log relative abundance. I am left to wonder if data analysis choices are driving these results.

Finally, I am puzzled as to why the authors chose to take a rigorous variance-based approach (ILR) and analyze it using a distance-based metric. It is established in the CoDa literature that Mahalanobis distances are often more appropriate for clustering based approaches and for outlier detection (see for example: http://www.statistik.tuwien.ac.at/public/filz/papers/2010Compstat.pdf). Did the authors examine other distances?

I believe that Figure 4 is somewhat over-interpreted. Subsection “Balance variance and microbiota assembly”, last paragraph: There are many technical reasons such as inefficient clustering, etc. that closely related sequences (called bacteria here) could exhibit strong correlations. Reading Lovell (referenced in this manuscript), and the points shown in 4G-L, indicate that an alternative explanation to that given in the text (see aforementioned subsection, end of second paragraph) is that a higher balance variance would be compatible with a model of indifference rather than competition since the vast majority of high variance ratio balances are likely to be non-correlated balances (Lovell 2015). Such non-correlated balances persist even out to the extreme tails and there is as yet no general principled approach to identify negative correlations in compositions.

Introduction, last paragraph: “the accuracy of distance.…” – this is a strong statement to make as the underlying standard of truth is assumed, not known from first principles.

Subsection “Addressing sparsity through weighting taxa”, first paragraph: I would argue that total read counts contain information on precision, not variance. It is hard to square this statement with the otherwise compositional approach used in the paper.

Subsection “Addressing sparsity through weighting taxa”, second paragraph: Why use a prior of one. The supplement that I found did not support the assertion that the prior chosen had little effect. This needs to be clarified.

Subsection “Human Microbiome Project (HMP)”: What is the effect of filtering and preprocessing on the method? As with many methods papers, the arbitrary choices here have the potential to become the accepted norm later without evidence.

Subsection “Supervised Classification”, first paragraph: Were the log-transformed data not scaled? if not, this is a problem since it is not a fair comparison.

---

## [Author Response]

*Essential revisions:*

*1) There was consensus among the reviewers that the normalization scheme seemed a bit ad-hoc and poorly justified. We suggest that it is made clear to the readers that the choice of weights used in the manuscript, both on the taxa and on the tips, are preliminary choices and further work will be required to ultimately determine the best weighting scheme. In addition, some further clarification on how sparsity was treated would strengthen the manuscript.*

We understand the reviewers’ concern regarding the discussion of our chosen weights (both branch length weights and taxa weights) and how sparsity was handled. We also recognize that these chosen weights are heuristics motivated in part on benchmarking results, and that further work is needed to determine optimal weightings in different situations. In response to these comments, we have expanded our discussion of these issues in the Results, Discussion and Methods sections (see subsection “Constructing the PhILR transform”; Discussion, sixth paragraph; subsections “Soft thresholding through weighting taxa”, “Incorporating branch lengths” and “Implementation”). In particular, we now state in the main text:

“Both the taxa weights and the branch lengths weights we introduce here may be viewed as preliminary heuristics; future work will likely yield additional weighting schemes, as well as knowledge for when a given weighting scheme should be matched to an analysis task.”

We have also included an additional supplementary figure (Figure 4—figure supplement 2) to provide readers with intuition for our chosen taxa weights behave in the face of data sparsity; namely, that these weights assign smaller influence to taxa with more zero and near-zero counts.

*2) None of the reviewers were entirely satisfied with the demonstration that the proposed transformation was superior to other transformations. It is unclear what it means, for example, that the PhILR technique sometimes produces a stronger clustering signal and sometimes a weaker clustering signal than weighted UNIFRAC (Figure 2). There was some discussion among the reviewers about whether a comparison to random binary partitions would be appropriate since this could reflect your null model (that phylogenetic binary partition is superior). Reviewer #2 was not enthusiastic about this idea and argued that a change on a tree does not alter (Aitchison) distances between data points. Therefore, results would likely not change no matter which tree has been selected for defining ILR coordinates.*

*In the end, the consensus of the reviewers was that exploring such null models might be of interest, but would not be appropriate as a requirement for revision. Nonetheless, reviewer #1 would have been interested to see if machine learning algorithms in fact performed better on the phylogenetic partition in comparison to random partitions. We leave it up to the authors as to whether they feel such an inclusion would improve their paper or be worth the effort. However, all reviewers felt that paper would be improved by more discussion of how to tell when one normalization procedure is superior to another. It seemed to reviewer #1 that it might be adequate to say that the proposed transformation appears to be doing no harm relative to weighted Unifrac but was providing some compositional safeguards. Reviewer #2 argued that the main benefit of the proposed procedure was in interpretability in yielding a phylogenetically aware transformation that could be utilized for test-statistics.*

We recognize from the reviewers’ comments that our rationale for providing the benchmarking results could be improved. Our goal in Figure 2 was to demonstrate that common microbiota analyses could be easily performed on PhILR transformed data with performance on par with current practice. Importantly, we did not intend these benchmarks to justify a compositionally robust approach to microbiota data analysis; we believe that this need has been well established (1-7). Nor do we believe it essential to our manuscript that PhILR outperforms other transforms or that Euclidean distance measured on PhILR transformed data outperforms all other (non-CoDA) distance metrics. In fact, we find it extremely unlikely that PhILR would always outperform any other transformation or distance metric under all situations. For example, while Weighted UniFrac may not be grounded in compositional data analysis theory, it was specifically designed for analysis of mixed microbial communities and we therefore would expect it to preform quite well in these tasks. We do think though that the results of Figure 2 are promising and suggest that at least with respect to our chosen metrics, PhILR transformed data can improve the performance of select tools in a variety of situations.

To reflect the reviewers’ overall concerns and our rationale for benchmarking, we have reworded our description of our benchmarking tasks and results throughout the manuscript. For example, we now motivate our use of benchmarking as follows:

“[…] we underscore that we do not find it essential that PhILR demonstrates superior benchmark performance to motivate its use in microbiota analysis. We believe that the need for compositionally robust tools has already been well established (1-7) and intended these benchmarks to showcase the flexibility and utility of working with PhILR transformed data.”

Additionally, we have been more explicit regarding the metrics used to evaluate the performance of PhILR transformed data. For example, we now specify in the text that the R^2^ statistic from PERMANOVA is used in Figure 2 and classification accuracy is used in Figure 2.

With regards to tests involving random binary partitions, we strongly agree with reviewer #2's comments. In the absence of branch length weights, all ILR transforms would be expected to preform equally well on our benchmark tasks. In addition, it is not our intention to show that PhILR exhibits superiority to other ILR transforms with regards to benchmark performance. Rather, we believe much of PhILR’s utility is that, by choosing the phylogeny as the partition, the resulting coordinates have biological meaning. Figure 3 and Figure 4 demonstrate how such meaning enables analyses involving evolutionarily relevant balances. Still, because the transformed space can be easily scaled by the phylogenetic distances between taxa (our branch length weighting), we believe the most natural “null” comparison to include is between PhILR with and without branch length weights. We include such an analysis in [Supplementary-material SD4-data], which demonstrates that, when coupled with our taxa weights, our chosen branch length weights had the highest rank in 9 of the 12 supervised classifications tasks and 2 of the three distance based tasks. We have updated our Discussion section to reflect the relation between the PhILR transform and other ILR transforms. These updated lines now include the statement:

“We also note that for some analyses, a phylogeny-based ILR transform will not outperform an ILR transform built from another sequential binary partition. […] Instead, what distinguishes the PhILR transform from other ILR transforms is the interpretability of the transformed coordinates. Balances in PhILR space correspond to speciation events, which can be both a source and target for biological insight.”

*3) There was general consensus among the reviewers that Figure 4 was unconvincing in that technical (non-biological) features of the algorithm could explain the results more easily than exclusionary dynamics. The authors could likely remove Figure 4 without damaging the paper, but if they choose to keep it, they should respond to the concerns below from reviewers #1 and #3 as both reviewers felt that simple lack of association between taxa and stochastic variation in sampling could explain many of the observations in Figure 4.*

Based on the reviewers’ astute comments, we have dramatically narrowed our ecological interpretation of Figure 4. Notably, because we now recognize that high balance variance can be due to either a lack of association or opposing dynamics between neighboring clades, we no longer draw conclusions regarding the importance of competitive exclusion in structuring microbial communities. We have chosen to instead restrict our discussion of Figure 4 to the separate conclusion that more phylogenetically similar clades are more likely to covary within human body sites.

In addition, to address reviewers’ concerns about technical sources of bias, we have also added a new paragraph in to the Results section which discusses the steps we took to avoid artifact in our analysis (subsection “Balance variance and microbiota assembly”, second paragraph). (We believe this new paragraph also has the benefit of reducing confusion regarding how our zero handling and the use of weights differs from prior sections.) We believe one of the steps to avoid bias that we now detail, which is a permutation-based hypothesis test, accounts for many of the potential technical artifacts that the reviewers alluded to. Histograms from the null distributions for each body site (Figure 4—figure supplement 3) are symmetric about zero, suggesting that any bias introduced by our handing of zero values or stochastic sampling was likely minimal. Still, we do acknowledge that we cannot completely exclude the possibility that inefficient clustering of sequences into OTUs could add to the observed pattern between balance variance and phylogenetic depth. But, due to our observation that this relation appears to be stronger above the species level than below, we believe that inefficient clustering alone could not explain the pattern we observed. We have added a discussion of this issue to our Discussion section (fifth paragraph).

Ultimately, we believe the variance analyses described in this section to be technically novel and uniquely enabled by PhILR. Even if future work were to determine our ecological conclusions dubious, we believe it a service to the community to highlight new analytical approaches that could eventually prove useful when interpreted in some other manner. Given this novelty, as well as the measures we took to avoid artifact, we therefore elected to retain Figure 4 (albeit in the setting of a more careful discussion). Still, if the reviewers still have strong reservations, we will gladly remove this section from our final manuscript.

*4) There was consensus among the reviewers that the authors could provide more guidance to the non-specialists about when this algorithm should be used. As reviewer #3 put it, when and why should a reader choose the PhILR approach and when should it be avoided.*

We appreciate this suggestion and have expanded on when the PhILR transform should be used and its limitations in our Discussion (last paragraph): “We foresee the PhILR transform being used as a default transformation prior to many microbiota analyses, particularly if a phylogenetic perspective is desired. For example, the PhILR transform could be used in lieu of the conventional log transform, which is often the default choice in microbiota analysis but not robust to compositional effects.”

Based on comments of reviewer #3 we have also added a Discussion paragraph discussing inference in terms of the changes of the relative abundance of individual taxa rather than balance values (seventh paragraph). This paragraph points out how single taxon analyses can actually be carried out after the ILR transform, by applying an inverse transform to model results.

*5) The authors should be explicit about what filtering and other pre-processing steps were used and whether those choices substantially impacted the performance of their algorithm.*

We agree with reviewers that these steps are important components of our analysis. Based on reviewer comments we have entirely rewritten the section of our Methods entitled ‘Datasets and Preprocessing’ in an attempt to clarify the preprocessing steps taken. In addition, we have created a new supplementary figure (Figure 2—figure supplement 2) to visually illustrate how our taxon weighting scheme assigns weights to taxa based on data sparsity.

We have also designed and performed new benchmarking analyses showing how data filtering choices affect model output ([Supplementary-material SD5-data]). For this analysis we redid our benchmarking results from Figure 2 for the CSS dataset under 4 different OTU filtering strategies. Notably these results include the performance of the other distance / dissimilarity measures we compared and the Log transformed and Raw relative abundance data. The four filtering strategies were as follows:

1) Keep taxa seen with at least 10 counts across all samples (this was the settings used in Figure 2);

2) Keep taxa that are seen with at least 2 counts in more than 1% of samples (settings used by McMurdie and Holmes (8));

3) Keep all taxa that represent at least 0.005% of total reads (settings suggested by Qiime pipeline (9));

4) Keep all taxa that are at least 0.1% abundant in any sample (settings used by Gloor and Reid (10)).

The resulting 4 versions of the CSS dataset varied in the number of OTUs remaining by 16-fold (minimum of 366 and maximum of 6142). In spite of this wide variation, we found PhILR’s results to be robust ([Supplementary-material SD5-data]).

*Reviewer #1:*

*[…] Suggestions for improvement follows:*

*1) The mathematical description of the algorithm could be improved in terms of its clarity. For example, it is not always clear how the pluses and minuses in the second equation of the subsection “The ILR Transform” are defined. It is not made immediately clear that the leftmost term (sqrt(ni+ * ni- / (ni+ + ni-)) in that equation is a correction for branch length. For the equation in the subsection “Addressing sparsity through weighting taxa”, please use parentheses to emphasize that the formula is sqrt(sum(xj)) * ((sum(cj 1…N)) ^ 1/N and not sqrt(sum(xj))^N * sqrt(sum(cj 1…N))*

Based on reviewer comments we have made numerous revisions aimed at improving the clarity of the Methods section entitled ‘The ILR Transform’. We do note that the term *(sqrt(ni+ * ni- / (ni+ + ni-))* is not a correction for branch length but is the scaling term that ensures the ILR basis elements have unit length. This term can be thought of as a scaling factor that ensures that balances with different numbers of descendant tips are still comparable on the same scale. We have attempted to clarify this point in our revised manuscript. We have also rearranged the equation for taxa weights in the section entitled ‘Soft thresholding through weighting taxa’ to avoid potential confusion pointed out by the reviewer.

*2) A limitation of working in ratio space in 16S data is what to do with all of the zeros in the spreadsheet (the "sparse data" problem). There is nothing inherent to the balancing algorithm that can deal with this problem (the sum in the denominator of the balance obviously can't be zero). The authors therefore require some kind of pseudo-count scheme and they have settled on one (given in the equation in subsection “Addressing sparsity through weighting taxa”) which is a function of the number of samples. If I am reading their formula correctly, the weight given to a taxa with 1 read in one sample and zero in all others would be very close to the sqrt(number of samples). The authors might consider giving this as an example and explicitly noting that their method has some similarity to adding the sqrt(number of samples) as a pseudo-count.*

We regret our lack of clarity regarding how sparsity was handled. Please note that the scheme referenced by the reviewer is our ‘taxa weighting’ strategy. This strategy is meant to be independent of how zero-replacement is handled (in our case, we add a pseudo count of 1). We hope the reviewer will find the new description of our taxa weights and sparsity to be clearer (as referenced in response to Essential revision 1). We have also added a new figure (Figure 2—figure supplement 2), which illustrates the weights assigned by our ‘taxa weighting’ to individual taxa.

*3) I struggled a bit with the variance analysis which concludes the Results sections. In the second paragraph of the subsection 2Balance variance and microbiota assembly” it is argued that "When the variance of a balance between two clades approaches zero, the mean abundance of taxa in each of the two clades will be linearly related and thus exhibit shared dynamics across microbial habitats". While this is true (and must be true mathematically as when the variance approaches zero, the ratio of the two normalized abundances of the clades is constant, which means they will become perfectly linearly related with a positive slope), it's not clear to me that a high variance necessarily means "exclusionary dynamics across samples". I don't see how a high variance in this number indicated anything about how direct or indirect the interactions are within the tree. There are lots of reasons two clades could lack perfect linear correlation with a positive slope from direct competition to being completely unrelated biologically with abundances just shifting due to stochastic variation. Also, wouldn't variance have to be lower closer to the tips of the phylogenetic tree as the number of sequences is lower, so the pseudo-count has more of an effect depressing overall variance? Doesn't that explain Figure 4? And if I am reading these graphs correctly, below the species level there are not enough datapoints to support modeling via localized regression one way or another. Although not central to the arguments of the paper, I found Figure 4 to be less convincing than other sections of the paper.*

We found these concerns helpful and valid, and we address them more fully in our response above to Essential revision 3. In short, based on these comments, we have dramatically narrowed our ecological interpretation of Figure 4 and more clearly described our multiple efforts to avoid potential artifacts of analysis. We also shared the concern that variance would be lower closer to the tips due to the heightened effects of pseudocounts; we therefore did not use pseudocounts for this analysis and instead conditioned our analysis on non-zero counts while more strictly excluded taxa with many zeros (to avoid analyses involving pseudocounts). We also incorporated a permutation-based hypothesis test, which should account for biases associated with count variation.

*Reviewer #2:*

*The manuscript introduces two new powerful ideas. They are: (a)construction of the sequential binary partition used to define*

*ILR coordinates on the base of phylogenetic trees; (b) use therecently published balances in weighted compositions to account forphylogenetic distances between OTU's or taxa. Both ideas aredeveloped and applied successfully to benchmark data sets.*

*For the originality of main ideas applied to microbiome analysis,the manuscript deserves publication.*

We thank the reviewer for these comments. On a minor note, we clarify that we do not use the weighted ILR transform as a means for introducing the phylogenetic distances between OTUs. Rather, we use a weighted transform to down-weight the effects of taxa with many zero and near-zero counts. We refer to this as taxa weighting. To account for phylogenetic distances, we simply scale balances by the phylogenetic distance between descendant clades. We do acknowledge that there is a deeper connection between these two sets of weights; that is, there is a close relationship between linearly scaling an ILR balance and changing the reference measure of the simplex (see (11), equation 18 in Appendix B); but for simplicity, we have not highlighted this connection in this manuscript. We hope that the updated description of these weightings in both the Results and Methods sections will make this distinction clearer.

*Reviewer #3:*

*The manuscript is, in general, written in a clear and logical manner with a minimum of technical jargon. Figure 1, Figure 3 and Figure 4 are outstanding examples of data visualization and demonstrate the utility of this approach over others extant in the field. Figure 5 seems an afterthought and could be combined with Figure 1, or made supplementary. The binary portion could also be completely described in the Methods without a figure.*

We agree that Figure 5 is much simpler than Figure 1–Figure 4 and wished to make it a supplementary figure. However, formatting constraints would require making it supplementary to a parent figure. The most natural parent figure would be Figure 1, but we believe that Figure 5’s sign matrix representation of a binary tree would confuse readers at that point in the manuscript. For this reason, we have chosen to leave Figure 5 in its current place.

*Figure 2, and the associated descriptive text are somewhat underwhelming demonstrations of the utility of the approach, which, in theory, should clearly outperform all other methods if the compositional data approach is necessary. The use of box-plots probably does not aid the visual appeal.*

*That the ILR based distance can seriously underperform the weighted unifrac metric is puzzling and cause for concern, since PhILR becomes another metric to be 'mined' to achieve the desired outcome.*

As we describe more fully in our response to Essential revision 2 above, we do not agree that PhILR must always outperform other methods to motivate a compositional data approach to microbiome data. First, we argue that the need for compositionally robust tools for microbiome data analysis is well established (1-7). Second, we include these benchmarks to assuage any concerns that PhILR is somehow not compatible with desired downstream analyses, like ordination analysis or supervised learning. Third, we find it extremely unlikely that PhILR would always outperform any other method on all experiments. This is especially true when comparing to tools tested and specifically designed for microbiota analysis such as Weighted UniFrac.

*Does the PhILR partitioning scheme outperform a simple random binary partition? this is crucial to know.*

As we state above in Essential revision 2, we strongly agree with reviewer #2’s comments. In the absence of branch length weights, all ILR transforms should perform equally. Moreover, we believe that much of PhILR’s utility derives from casting the resulting coordinates in the evolutionary framework afforded by use of a phylogenetic tree.

*There needs to be serious guidance given to the reader as to when and why to choose the PhILR approach and when not to. The same can be said for the classification methods shown, where the PhILR generally (but not always) outperforms relative abundance, but not log relative abundance. I am left to wonder if data analysis choices are driving these results.*

We appreciate this guidance and have inserted a statement into the Discussion suggesting the use of PhILR as a default microbiota transformation, prior to further analyses, particularly when a phylogenetic perspective is desired (see Essential revision 4 for more details). Still, we do feel that an in-depth discussion of classification methods is beyond the scope of this work. Such choices are ultimately data driven and the subject of extensive research in the Machine Learning community. Our rationale for choosing 4 separate classification algorithms (both parametric and non-parametric) was simply to demonstrate that a range of algorithms can be applied to PhILR transformed data.

*Finally, I am puzzled as to why the authors chose to take a rigorous variance-based approach (ILR) and analyze it using a distance-based metric. It is established in the CoDa literature that Mahalanobis distances are often more appropriate for clustering based approaches and for outlier detection (see for example: http://www.statistik.tuwien.ac.at/public/filz/papers/2010Compstat.pdf). Did the authors examine other distances?*

While the Mahalanobis distance is a very useful tool for CoDa (and data analysis in general), we believe its use is neither simple nor standard practice. We therefore feel it more appropriate to leave this as a potential improvement on the PhILR method for cases where outliers are a chief concern.

*I believe that Figure 4 is somewhat over-interpreted. Subsection “Balance variance and microbiota assembly”, last paragraph: There are many technical reasons such as inefficient clustering, etc. that closely related sequences (called bacteria here) could exhibit strong correlations. Reading Lovell (referenced in this manuscript), and the points shown in 4G-L, indicate that an alternative explanation to that given in the text (see aforementioned subsection, end of second paragraph) is that a higher balance variance would be compatible with a model of indifference rather than competition since the vast majority of high variance ratio balances are likely to be non-correlated balances (Lovell 2015). Such non-correlated balances persist even out to the extreme tails and there is as yet no general principled approach to identify negative correlations in compositions.*

We thank the reviewer for this astute observation and have removed our ecological interpretation of higher balance variances. Please see our above response to Essential revision #3 for additional details.

*Introduction, last paragraph: “the accuracy of distance…” – this is a strong statement to make as the underlying standard of truth is assumed, not known from first principles.*

Thank you for pointing this out. We agree it was an overly strong statement and in our revised draft, we have softened it to read:

“the accuracy of supervised classification methods on our benchmark datasets was matched or improved with PhILR transformed data”

*Subsection “Addressing sparsity through weighting taxa”, first paragraph: I would argue that total read counts contain information on precision, not variance. It is hard to square this statement with the otherwise compositional approach used in the paper.*

We have updated our discussion of this issue throughout the paper to reference precision rather than variance. Nevertheless, although we are of course strong proponents of compositional approaches, we do believe that including uncertainty from count data could lead to exciting advances in the analysis of microbiota data, which we briefly describe in our revised Discussion (sixth paragraph).

*Subsection “Addressing sparsity through weighting taxa”, second paragraph: Why use a prior of one. The supplement that I found did not support the assertion that the prior chosen had little effect. This needs to be clarified.*

We regret the lack of clarity in this area. The equation referenced represents our strategy for how taxa weights were chosen. This strategy is meant to be independent of how zero-replacement is handled (in our case, we add a pseudo-count of 1). We acknowledge our choice of pseudo-count is a heuristic, which was supported largely by our benchmarking results ([Supplementary-material SD4-data]) and does not come with any optimality guarantees. We did investigate how changing the pseudo-count added to the table affected our benchmarking results. Changing this value from 1 to 2, 3, or 10 had minimal effect (also [Supplementary-material SD4-data]). We have tried to clarify our description of taxa weights as well as our handling of zero values in the revised manuscript (see our response to Essential revision 1, above).

*Subsection “Human Microbiome Project (HMP)”: What is the effect of filtering and preprocessing on the method? As with many methods papers, the arbitrary choices here have the potential to become the accepted norm later without evidence.*

We understand this concern and now tested the effects of alternate filtering and preprocessing schemes on our method. In short, the effects are minimal on PhILR. Please see our response to Essential revision 5 above for more detail.

*Subsection “Supervised Classification”, first paragraph: Were the log-transformed data not scaled? if not, this is a problem since it is not a fair comparison.*

To preface, we understand this comment to be asking whether the log transformed data were scaled in the sense of how scaling is often applied in machine learning; that is, each variable is often scaled to a unit normal distribution or to exist between 0 and 1. If indeed this is what the reviewer was asking, then no, the log-transformed data were not scaled. But, then again, according to this criterion, the PhILR transformed data were also not scaled.

Alternatively, if the reviewer is referring to the scaling of the ILR basis elements that is part of the definition of the ILR transform (this is the ni+ni−/ni++ni− term in the transform), we are not aware of an obvious analogy between this and a potential scaling step for the log transform that is commonly used.

Bibliography:

1) Jackson DA (1997) Compositional data in community ecology: The paradigm or peril of proportions? Ecology 78(3):929-940.

2) Friedman J & Alm EJ (2012) Inferring correlation networks from genomic survey data. PLoS Comput Biol 8(9):e1002687.

3) Aitchison J (1986) The statistical analysis of compositional data (Chapman and Hall, London; New York).

4) Lovell D, Müller W, Taylor J, Zwart A, & Helliwell C (2011) Proportions, percentages, ppm: do the molecular biosciences treat compositional data right. Compositional Data Analysis: Theory and Applications, eds Pawlowsky-Glahn V & Buccianti A (John Wiley & Sons, Ltd.), pp 193-207.

5) Gloor GB, Macklaim JM, Vu M, & Fernandes AD (2016) Compositional uncertainty should not be ignored in high-throughput sequencing data analysis. Austrian Journal of Statistics 45(4):73.

6) Li HZ (2015) Microbiome, Metagenomics, and High-Dimensional Compositional Data Analysis. Annual Review of Statistics and Its Application, Vol 2 2(1):73-94.

7) Tsilimigras MC & Fodor AA (2016) Compositional data analysis of the microbiome: fundamentals, tools, and challenges. Ann Epidemiol 26(5):330-335.

8) McMurdie PJ & Holmes S (2013) phyloseq: an R package for reproducible interactive analysis and graphics of microbiome census data. PLoS One 8(4):e61217.

9) Navas-Molina JA, et al. (2013) Advancing our understanding of the human microbiome using QIIME. Methods Enzymol 531:371-444.

10) Gloor GB & Reid G (2016) Compositional analysis: a valid approach to analyze microbiome high-throughput sequencing data. Can J Microbiol 62(8):692-703.

11) Egozcue JJ & Pawlowsky-Glahn V (2016) Changing the Reference Measure in the Simplex and its Weightings Effects. Austrian Journal of Statistics 45(4):25-44.